

# Operational algorithm for ice/water classification on dual-polarized RADARSAT-2 images

Natalia Zakhvatkina[1,2], Anton Korosov[3], Stefan Muckenhuber[3], Stein Sandven[3], Mohamed Babiker[3]

[1]Nansen International Environmental and Remote Sensing Centre (Nansen Centre, NIERSC), 14th Line 7, Office 49, Vasilievsky Island, St. Petersburg, 199034, Russian Federation
[2]Arctic and Antarctic Research Institute (AARI), Bering str. 38, St. Petersburg, 199397, Russian Federation
[3]Nansen Environmental and Remote Sensing Center (NERSC), Thormøhlensgate 47, 5006 Bergen, Norway

*Correspondence to*: Natalia Zakhvatkina (natalia.piotrovskaya@niersc.spb.ru)

**Abstract.** Synthetic aperture radar (SAR) data from RADARSAT-2 (RS2) taken in dual-polarization mode provide additional information for discriminating sea ice and open water compared to single-polarization data. We have developed a fully automatic algorithm to distinguish between open water (rough / calm) and sea ice based on dual-polarized RS2 SAR images. Several technical problems inherent in RS2 data were solved on the pre-processing stage including thermal noise reduction in HV-polarization channel and correction of angular backscatter dependency on HH-polarization. Texture features are used as additional information for supervised image classification based on Support Vector Machines (SVM) approach. The main regions of interest are the ice-covered seas between Greenland and Franz Josef Land. The algorithm has been trained using 24 RS2 scenes acquired during winter months in 2011 and 2012, and validated against the manually derived ice chart product from the Norwegian Meteorological Institute. Between 2013 and 2015, 2705 RS2 scenes have been utilised for validation and the average classification accuracy has been found to be 91 ± 4 %.

## 1 Introduction

Synthetic aperture radar (SAR) is an active microwave sensor, which has the capability to acquire high resolution, large spatial coverage data through clouds independently of day or night. This is especially crucial for sensing polar regions, where SAR data are widely used for exploring sea ice concentration, extent changes, thickness distribution, detection of leads, polynyas, ice floes and ice edge, ice type identification and classification (Johannessen et al., 2007; Dierking, 2013). Monitoring of dynamic sea ice process in polar region, i.e. ice edge dynamic distribution and motion, is important for navigation and scientific tasks. High-resolution data from C-band SAR from ERS-1/2 (European Remote Sensing satellites, European Space Agency (ESA)), RADARSAT-1 (Earth observation satellite, Canadian Space Agency) and ENVISAT (Environmental Satellite, ESA) are often used as the main data source for sea ice monitoring (Johannessen et al., 2007). The advanced capabilities of SAR on board of RADARSAT-2 (RS2) and Sentinel-1 (European Commission and ESA) with multi-polarization options benefit the sea ice applications including improved sea ice dynamic observation, ice edge and ice types detection, and extend operational functionality.



The objective of sea ice classification on SAR images is to identify sea ice types and open water (OW) areas based on surface roughness and other characteristics of the scene. Methods based only on utilization of the backscattering coefficients (sigma0 or $\sigma°$) for discrimination between sea ice and open water are hampered by ambiguities in the relation between ice types and $\sigma°$, since various ice types (multiyear, first-year and some young and new ice) and open water depending on wind speed can have similar $\sigma°$ (Dierking, 2010; Johannessen et al., 2007). In particular, discrimination between open water during low wind (calm open water) and first-year ice, new ice or young ice with frost flowers and multiyear ice, can be problematic. Including additional image characteristics, like image texture and others, can improve the classification results significantly (Shokr, 1991; Soh and Tsatsoulis, 1999; Clausi, 2002; Bogdanov et al., 2005; Maillard et al., 2005; Yu et al., 2012).

Arctic sea ice concentration and polynya detection using ERS and RADARSAT-1 SAR data have been studied by Dokken et al. (2002). The SAR polynya detection algorithm introduced by Dokken et al. (2002) is based on wavelet transforms for edge detection and standard texture-based methods. A threshold function using texture information is used to classify sea ice and water for polynya detection. A semi-automated sea ice classification method based on fuzzy rules was presented in Gill (2003) classifying RADARSAT-1 data over the Arctic into calm water, wind-roughened water, and sea ice of low and high concentrations. Advanced Reasoning using Knowledge for Typing of Sea Ice (ARKTOS) (Soh et al., 2004) has been established to support scientific research and operational applications in the field of sea ice segmentation and classification. Norut IT has developed a fully automatic algorithm for ice/ocean discrimination in RADARSAT-1 and ENVISAT SAR imagery (Haarpainter and Solbø, 2007). The algorithm is texture based, consists of an automatically trained maximum likelihood classifier and divides the SAR images into slices of small incidence angle range. The results show that sea ice and water can be discriminated quite reliably. Some examples showed a tendency of the algorithm to a better performance at low incidence angles. Karvonen et al. (2005) distinguished the Baltic Sea ice from open water on SAR images based on thresholding of segment-wise local autocorrelation. The accuracy of this method was about 90 % compared to digital ice charts for the Baltic Sea. This algorithm has been used by the Finnish Meteorological Institute (FMI) and tests with RADARSAT-2 and ENVISAT SAR data show that over 89.4 % of the test data agree with ice classification provided by the Finnish Ice Service for the Baltic Sea and Arctic Sea (Karvonen, 2010, 2012).

Dual-polarization has several advantages for sea ice identification and classification compared to single-polarization SAR data. Gray ice and multiyear ice, while being very different in their thickness (10 – 15 cm and more than 2.5 m, respectively), show similar brightness in the HH channel, but MYI appears brighter than gray ice in the HV channel. New ice and wind roughened open water can both appear bright in HH, and are therefore difficult to distinguish in single polarization from the sea ice. However, new ice and open water, especially affected by wind, are darker in HV, providing information to improve sea ice classification (Sandven et al., 2008). The dual-polarization ENVISAT SAR Alternative Polarization Mode data show possibility of decision-tree classifier for sea ice types and open water discrimination using estimated statistical thresholds for winter. Open water is unambiguously discriminated (except thin sea ice) from smooth FYI, rough FYI, and MYI using a co-polarized ratio threshold with > 99% accuracy (Geldsetzer and Yackel, 2009). The





potential of supervised K-means and maximum likelihood classification of various SAR polarimetric data to three pre-identified sea ice types and wind-roughened open water is explored in Gill and Yackel (2012).

A MAp-Guided Sea Ice Classification System (MAGIC) has been developed by the Canadian Ice Service (CIS) as an automated ice-water discrimination system using dual polarization images from RADARSAT-2. MAGIC is a combination of

a "glocal" Iterative Region Growing using Semantics (IRGS) classification (Yu and Clausi, 2008) and with a pixel-based Support Vector Machine (SVM) approach. The result of the glocal classification is an identification of homogeneous regions with arbitrary class labels. The ice-water map is provided by the SVM classifier, exploiting SAR texture and backscatter features. The MAGIC system has been applied on 20 RS2 scenes over the Beaufort Sea. The average classification accuracy with respect to manually drawn ice charts is 96.5% (Clausi et al., 2010; Ochilov and Clausi, 2012; Leigh et al., 2014).

A Neural network (NN) based algorithm has been developed for ENVISAT SAR images for operational sea ice classification including validation (Zakhvatkina et al., 2013). The algorithm discriminated between level FYI, deformed FYI, MYI and open water/nilas in the high Arctic during winter conditions and showed good applicability in the Central Arctic. Using the same approach, an algorithm for mapping ice / water utilizing ENVISAT ASAR WSM images was created for operationally ice edge detection in Fram Strait. The ice / water classes were estimated by a multi-layer perceptron (MLP)

neural network which uses SAR calculated texture features and concentration data from AMSR (Advanced Microwave Scanning Radiometer) and, later, SSM/I (Special Sensor Microwave/Imager) as inputs (Sandven et al., 2012). Daily ice/water products were provided with a resolution of 525 m from winter 2011 until April 2012. The accuracy of this classification was about 97 % compared to high resolution sea ice concentration charts based on manual interpretation of satellite data provided by the Norwegian Meteorological Institute.

Our goal is to extend the ENVISAT single polarization method (Sandven et al., 2012) by utilizing dual polarization data from RS2 and to develop an algorithm for ice/water classification, which can be operationally applied to RS2 data for the production of ice/water maps as part of the GMES and Copernicus programme. The presented algorithm is based on texture features and SVM method using the advantages of dual-polarization RS2 SAR image data.

The paper describes the developed algorithm and discusses practical issues of applicability. The presented algorithm's

scheme with described steps and parameters of their realization may be implemented by users without additional complicated segmentation step. The paper is organized as follows: Sect. 2 introduces the used satellite images and considered area of interest. The algorithm including pre-processing and validation procedure is described in Sect. 3. The results of the pre-processing step, ice/water classification and comparison with manual ice charts are presented in Sect. 4. The discussion can be found in Sect. 5.

**2 Data**

The regions of interest are the ice-covered seas between Greenland and Franz Josef Land, where an updated analysis of the ice situation is particularly important due to a strong seasonal cycle of sea ice cover and highly variable sea ice export out of



the Arctic through Fram Strait (Vinje and Finnekåsa, 1986). Since optical sensors cannot provide useful data during cloud cover and polar night, SAR represents the only sensor capable of providing high-resolution data all year round.

This study is based on RS2 ScanSAR Wide (SCW) mode images with 500 km swath width, a pixel spacing of 50 × 50 m and dual-polarizations (HH+HV). This is the main mode used by RS2 for operational sea ice monitoring (RS2 Product Description, 2011). Twenty four SCW scenes around Svalbard (Fig. 1) from 2011 and 2012 were utilized in the following analysis for the training the algorithm. The images were selected during winter months to cover various types of thin ice (e.g. new and young ice), first-year and multiyear ice, characterized by different degrees of deformation, packed ice, broken ice and open water with different wind speed conditions (rough, very rough and calm water including leads and dark nilas).

The backscatter at HH generally decreases with increasing incidence angle (Fig.2a), whereas the HV channel is less sensitive to the incidence angle. HV is a mode of radar polarisation, where the microwaves of the electric field are oriented in the horizontal plane for signal transmission, and in vertical plane for signal receiving. The HV channel of SAR appears darker than the HH channel because the radar backscatter values of the cross polarization are generally rather low and close to (in some cases even below) the system noise level. The reason is that horizontally polarized transmitted pulses have a tendency to scatter backwards in horizontal polarization (as seen in HH) rather than vertical polarization (as seen in HV). This causes the HV channel to include disturbances in along-track direction (visible as bright and dark stripes), which occur because the ScanSAR Wide Beam mode assembles wide SAR image from several narrower SAR beams or burst boundaries (Fig. 2b). The expected noise level is a local mean noise power value that fluctuates across the image, and is obtained from a model that accounts for the characteristics of the payload, the beam mode, the acquisition, and the ground processing [RS2 PUG] (Jefferies, 2012). The system noise level as a function of the incidence angle is documented in the XML file that comes with the RS2 image.

## 3 Methodology

Our automated ice / water classification algorithm includes six main steps:

1) SAR data pre-processing including reduction of thermal noise effect for HV, incidence angular correction for HH, and absolute RS2 image calibration to obtain σ° values for both channels.

2) Manual classification of SAR images into predefined classes (e.g. ice and water or more classes if needed) under consideration of auxiliary information like optical data, ice concentration from passive microwave, previous classification results and historical data.

3) Calculation of texture features from HH and HV images.

4) Training of automatic classifier (e.g. SVM) for classification of arrays with some chosen texture features as well as σ° values based on the results of manual classification.

5) Application of automatic classifier to divide the RS2 scene into the predefined classes.

6) Validation of the classification results using manually drawn ice charts.



After completing the algorithm training, the fully automated image classification includes only three of the above mentioned steps: 1) reprocessing; 3) texture feature retrieval; and 5) application of the automatic classifier (SVM).

The initial size of the full resolution RS2 SCW image is c.a. 10000 × 10000 pixels. We downscale the original image by averaging to 5000 × 5000 pixels to increase the computational efficiency and decrease the influence of speckle noise. The image size is further reduced during the computation of the texture features by using a sliding window with 16 pixels step size. The image size of the final product is about 330 × 330 pixels with 1600 m pixel spacing. This reduction in resolution significantly increases the processing throughput and allows computing a classification result in less than 15 minutes.

Pre-processing of RS2 data was performed utilizing the open source Python toolbox NANSAT (Korosov et al., 2015), [https://github.com/nansencenter/nansat/wiki]. The texture extraction algorithm was created in the Python programming language. The scikit-learn open source was used to implement the SVM classification method [http://scikit-learn.org/stable/index.html].

### 3.1 Incidence angle correction for HH

All images are corrected to a reference angle of 35°, which represents the centre incidence angle, and allows analyzing SAR images without brightness amplification. The angular dependence compensation technique consists of: 1) calculation of the backscatter coefficient using image brightness according to the ESA absolute calibration formula; 2) backscatter recalculation to 35° incidence angle using a predefined calculated coefficient. The coefficient was defined by calculating the linear trend of the observed backscatter signal on several HH-polarized RS2 SCWA images of pack ice. The procedure is similar to the pre-processing of ENVISAT ASAR data in Zakhvatkina et al. (2013). The backscatter normalization to a pre-defined incidence angle allows obtaining homogenous image contrast across the swath over ice-covered areas.

### 3.2 Thermal noise correction for HV

The thermal noise reduction consists of three steps: 1) reading 100 noise values and corresponding incidence angles from the XML file; 2) interpolation of noise on a finer grid for each pixel; 3) subtraction of interpolated noise values from the backscatter values of the entire image.

Due to the incontinuity of the noise floor at the boundaries of the individual SAR beams and the low resolution of the provided noise values in the XML file (only 100 points for 500 km swath width), the noise correction may result in an erroneous substraction of a high noise floor from a low signal of the neighboring SAR beam and hence, yield negative values for $\sigma°$. To prevent such flaws, a 10 pixels wide stripe of data along the edge of the SAR beam is masked out and disabled for further analysis.

### 3.3 Manual classification

Manual classification has been done for images depicting several different sea ice types and ice-free areas containing both rough (caused by strong winds) and calm (low wind speeds) open water. Predominant subclasses, which must be reliable and



undertaken with good quality, were identified and chosen by sea ice experts through visual analysis of RS2 scenes and according to their previous expertise. The selected images did not contain homogeneous ice cover, since the mixing of different ice types, different degrees of deformation, cracks, ridges and leads prevents the trained classifier from identifying distinct features. The main class 'sea ice' was chosen to include the following subclasses: one subclass including young ice, first-year ice and multiyear ice; fast ice; and broken ice on the edge (border) mixed with ice-free areas (mostly found in the marginal ice zone). The class 'open water' includes the two subclasses open water with high and very high wind speed conditions and a third subclass that represents a mixture of calm open water, frazil ice, leads and nilas. For the final product, the subclasses are merged into the main classes 'sea ice' and 'open water', since the similarities between the subclasses are too high for a reliable discrimination without additional data.

## 3.4 Calculation of texture features

The calculation of texture features consists of the computation of the gray level co-occurence matrix (GLCM) using Eq. (1) and the calculation of texture features based on the GLCM (Equations 2 – 10). Considering the full range of possible brightness levels (e.g. 0 – 255 for 8 bit data) and a small window size, most GLCM elements would be zero and that would have a negative effect on the classification result. Therefore, we divide the full range into few intervals (quantization levels $K$). The GLCM is created for each direction $\theta$, where each cell $(i, j)$ is a measure of the relative frequency of two pixels occurrence with brightness $i$ and $j$ respectively, separated by a co-occurrence distance $d$. One may also say that the matrix element $P_{d,\theta}(i,j)$ is a measure of the second order statistical probability for changes between gray levels $i$ and $j$ at a particular displacement distance $d$ and at a particular angle (direction) $(\theta)$. The size of square GLCM is equal to number of quantized brightness levels K. The GLCM is averaged over four directions $\theta$ (0º, 45º, 90º, 135º) to account for possible rotation of the ice floes (Clausi, 2002).

$$S_{d,\theta}(i, j) = \frac{P_{d,\theta}(i,j)}{\sum_{i=1}^{K}\sum_{j=1}^{K}P_{d,\theta}(i,j)} \tag{1}$$

where $S_{d,\theta}$ – GLCM, $P_{d,\theta}$ – number of neighbor pixel pairs, $\theta$ – fixed vector directions (0º, 45º, 90º, 135º), $d$ – co-occurrence distance, $K$ – number of quantized gray levels, $i, j$ - gray levels (0 – 255).

$$Energy = \sum_{i=1}^{K}\sum_{j=1}^{K}\left[S_{d,\theta}(i,j)\right]^2, \tag{2}$$

$$Homogeneity = \sum_{i=1}^{K}\sum_{j=1}^{K}\frac{S_{d,\theta}(i,j)}{1+(i-j)^2}, \tag{3}$$

$$Contrast = \sum_{i=1}^{K}\sum_{j=1}^{K}(i-j)^2 S_{d,\theta}(i,j), \tag{4}$$

$$Correlation = \frac{\sum_{i=1}^{K}\sum_{j=1}^{K}(i-\mu_x)(j-\mu_y)S_{d,\theta}(i,j)}{\sigma_x\sigma_y}, \tag{5}$$

$$Skewness = \sum_{i=1}^{K}\sum_{j=1}^{K}\frac{(S_{d,\theta}-\mu)^3}{\sigma^3}, \tag{6}$$

$$Kurtosis = \sum_{i=1}^{K}\sum_{j=1}^{K}\frac{(S_{d,\theta}-\mu)^4}{\sigma^4}, \tag{7}$$

$$ClusterShade = \sum_{i=1}^{K}\sum_{j=1}^{K}(i+j-\mu_x-\mu_y)^3 S_{d,\theta}(i,j), \tag{8}$$

$$Entropy = -\sum_{i=1}^{K}\sum_{j=1}^{K}S_{d,\theta}(i,j)log_{10}S_{d,\theta}(i,j), \tag{9}$$

$$ClusterProminence = \sum_{i=1}^{K}\sum_{j=1}^{K}(i+j-\mu_x-\mu_y)^4 S_{d,\theta}(i,j), \tag{10}$$



where $\quad \sigma_x{}^2 = \sum_{i=1}^{K}\sum_{j=1}^{K}(j-\mu_x)^2 S_{d,\theta}(i,j)\quad$ and $\quad \sigma_y{}^2 = \sum_{i=1}^{K}\sum_{j=1}^{K}(j-\mu_y)^2 S_{d,\theta}(i,j)\quad$ are standard deviation of rows and columns;

$\mu_x = \sum_{i=1}^{K}\sum_{j=1}^{K} i S_{d,\theta}\quad$ and $\quad \mu_y = \sum_{i=1}^{K}\sum_{j=1}^{K} j S_{d,\theta}\quad$ are mean values of rows and columns; $\quad \sigma^2 = \sum_{i=1}^{K}(i-\mu)^2 \sum_{j=1}^{K} S_{d,\alpha}(i,j)\quad$ - standard deviation

and $\mu = \sum_{i=1}^{K}\sum_{j=1}^{K} i S_{d,\alpha}(i,j)$ - mean values of brightness.

The results of this procedure depend on several factors such as the size of the sliding window, the co-occurrence distance,

and the quantization levels (Shokr, 1991; Soh and Tsatsoulis, 1999; Clausi, 2002). In order to test the effects of these parameters on the classification accuracy, texture features were calculated for the window sizes 16, 32, 64, and 128 pixels using different co-occurrence distances and varying the number of quantized gray levels from 16 to 64. The optimal values for the parameters of texture features calculation were selected.

### 3.5 Support Vector Machines

The Support Vector Machines are supervised learning methods with associated learning algorithms that provide data classification. The basic SVM takes a set of input data and predicts, for each given input, which of two possible classes forms the output, making it a non-probabilistic classifier. The divided into classes objects are mapped so that the examples of the separate categories are separated by a clear gap that is as wide as possible. SVM can also perform a non-linear classification using the kernel trick, implicitly mapping their inputs into high-dimensional feature spaces. A special property

of the SVM is that they simultaneously minimize the empirical classification error and maximize the geometric margin; hence, they are also known as maximum margin classifiers. The distance between the lines that separate classes is called the margin. The vectors (points) that constrain the width of the margin are the support vectors (Cortes and Vapnik, 1995).

The simplest way to divide two groups is a straight line, flat plane or an N-dimensional hyperplane. If groups can only be separated by a nonlinear boundary, SVM applies a kernel function to map the data into a different space where a hyperplane

can be used for the separation of the two groups.

The kernel function may transform the data into a higher dimensional space to make this separation possible. In our study we have used the radial basis function kernel (RBF kernel), which is found to work well in a wide variety of applications. It uses squared Euclidean distance between the two feature vectors and can be interpreted as a similarity measure.

The calculated texture features and $\sigma°$ values corresponding to the manually identified classes on several pre-processed RS2

images were used as input data for training the SVM classifier. After completing the training procedure, the resulting SVM can be used for automatic sea ice classification.



### 3.6 Validation

The validation of Arctic sea ice products is a challenging task due to the lack of ground truth data. As a substitute, our sea ice classification results have been compared with manual sea ice charts produced by the operational ice charting service at the Norwegian Meteorological Institute (MET Norway). MET Norway ice charts are produced every working day by sea ice

analysts using the following data sources: high resolution SAR images, low resolution microwave SSM/I and SSMIS data (DMSP), MODIS images (Terra and Aqua) and AVHRR data from NOAA. In our comparison, MET Norway ice charts are assumed to represent correct classification and the confusion matrix was calculated for accuracy evaluation of our algorithm results.

### 4 Results

To illustrate the algorithm performance, Figure 6 depicts the manual classification from sea ice expert analysis compared to the automatic SVM classification result based on the RS2 scene shown in Figure 2. The example scene was acquired on November 28, 2011 over the western part of Svalbard in Fram Strait. Figure 2 and 3 show both HH and HV polarizations before and after corresponding corrections described in Sect. 2: compensation of incidence angle effects for HH (Fig. 3a), and noise reduction for HV (Fig. 3b). The image contains several ice types, open water under different wind conditions and

land. The open water area is located on the right part of the image and increases from top right to bottom left. The sea ice area includes a marginal ice zone with bright broken up ice. The ice-covered areas and the rough OW areas appear both bright in HH and are therefore difficult to distinguish. Including HV, however, provides additional information, since OW areas appear generally darker than sea ice in HV. This is one of the major dual polarization advantages and can be seen in the lower right part of the example image (Figure 2).

**4.1 Correction for incidence angle and thermal noise**

The linear trend coefficient used for backscatter angular dependence correction of HH was estimated to be 0.298 and allowed normalization of $\sigma°$ to the incidence angle 35º as shown on Figure 3 a and c. The application of our noise correction procedure for HV reduces significantly thermal noise and get rid of vertical striping as shown in Fig. 3b, d.

**4.2 Texture features calculation**

As part of the algorithm development, texture features were calculated based on different parameter settings. The best results were achieved using the following parameter set: number of gray levels $K = 32$, co-occurrence distance $d = 8$, sliding window size $ws = 64 \times 64$, moving step of the sliding window $s = 16$. Using the following texture features for the two channels provided the best test results: HH - energy, inertia, cluster prominence, entropy, 3rd statistical moment of brightness, backscatter, and standard deviation; HV - energy, correlation, homogeneity, entropy, and backscatter. Including more texture

features for both channels was tested, but found not to improve the information content. The calculation parameters were





found experimentally to give a good compromise between speckle noise reduction, preservation of details and correct classification results [methodology description in Zakhvatkina et al., 2013].

Texture characteristics provide a more complete delineation of surface parameters than the raw backscatter signal, and increase the potential for ice and water separation. The scatterplots in Fig. 4 show the values of two different texture features plotted against each other and illustrate the usefulness of texture features for discrimination between defined classes.

### 4.3 Manual versus automatic classification

As described in Sect. 3, several SAR images were classified manually as part of the training procedure for the automatic algorithm. Comparing the manual classification with the algorithm results (Figure 6) reveals a general high level of correspondence and illustrates the capability of the automatic approach. Detailed observation of the classification results show that most misclassifications are observed near land and in the MIZ. Fig. 6b shows small features inside ice-covered zone (blue dots) that were misclassified as OW.

### 4.4 Validation

Validation of the algorithm results has been performed using 2705 RS2 images taken over our area of interest in the period 1 January 2013 until 25 October 2015. For each RS2 image, an error matrix based on pixel-by-pixel difference between algorithm result and MET Norway chart has been calculated. OW and sea ice correspondence as well as an overall accuracy were obtained for each RS2 image classification result and averaged accuracies have been calculated for each month. The impact of each class to the classification error has been estimated and the respective monthly averaged errors were computed. The averaged overall accuracies including standard deviation, and errors in ice and water classification for each month are given in Table I. In addition, the monthly accuracies are presented as a graph in Fig. 7. The monthly averaged overall accuracies show lower values during summer months (Fig. 7 - from May to October) and higher values during winter. The average total classification accuracy for all 2705 scenes is 91 ± 4 %.

Figure 8 shows an example of the validation process. The RS2 HH image is shown in Figure 8a, the result of our SVM classification in Figure 8c and the MET Norway sea ice chart in Figure 8b. To compare the algorithm result with the manually derived ice chart of the MET Norway service, both products are reclassified into ice and water (Figure 8d and 8e). The error matrix is represented as an image (Figure 8f) with the following three classes: no difference; sea ice error (sea ice at MET Norway, OW at our results); OW error (OW at MET Norway, sea ice at our results). The comparison of obtained results (Fig. 8f) of the automated classification (Fig. 8d) with ice charts created by MET Norway ice experts (Fig. 8e) showed that the use of the SVM gives more detailed ice cover distribution maps. The general MET Norway ice charts obtained manually using semi-automated technology; however, the fully automated classification provides a significantly higher rate of thematic processing.



## 5 Discussion

### 5.1 Incidence angle variations significance

Water areas depending on wind speed have a very large range of brightness and in general include backscatter values of all ice types. The higher wind reduces the contrast between open water and sea ice, which gives an ambiguity of these classes.

Strong dependence of the backscatter on incidence angle in open water surface is well known (Shokr, 2009) and is significantly higher than for sea ice, and the correction factor is very different for ice and water. It is generally not possible to eliminate the incidence angle effect applying a "water" angular correction to the scene with a mixture of various types of open water and sea ice when sea ice area tends to be smoother and has overestimated signatures. Since type of the surface is not known beforehand, we apply the correction for sea ice as default because ice cover surface has more reliable backscatter

ranges for various ice types for the subsequent classification. However, the total compensation is impossible as the backscatter dependence on the incidence angle varies for different ice types (Makynen et al, 2002) and water areas on the one scene. The radiometric corrections during calibration process are just a first-order approximation; nevertheless, the advantages of performing the angular correction are greater than the disadvantages (Moen et al., 2015). With regards to thermal noise correction, we can observe that sometimes not all visible noise floor artifacts inside beams can be completely

removed and these residuals may cause classification errors.

### 5.2 Number of texture features vs efficiency

In addition to the 9 extracted texture features we characterize the surface by values of $\sigma°$ averaged within the sliding window and a value of standard deviations. Given that we have two channels (HH and HV) the number of parameters grow up to 20 and some of them are strongly intercorrelated (Albregtsen, 2008). This similarity can explain the misclassifications and in

fact this is part of the motivation to reduce dimensionality. If we include too few texture features to the classifier then the poor features have to be compensated by using complicated discrimination function and can lead to increased classification confusion. On the other hand, if all texture features will be used by the classifier, some classes can be underestimated or overestimated and the discrimination for many classes may lead to higher classification errors. Selection procedure should be applied using a set of training image regions to establish the set of features and its computation parameters giving the

smallest classification error. Thus, we use the limited set of texture characteristics that provides a good classification with a small computational load. In other words, we constrain the texture features number by the demanded balance considering the SAR image level of details, computation time and the optimal reliable class separation.

### 5.3 Meaning and value of texture features

Each texture features describes some of the variability of pixel values within a sliding window and is detailed below.

Energy or Angular Second Moment expresses the uniformity or smoothness of SAR images, i.e. repeatability describes pairs of pixels that belong to the same range of brightness levels. Energy increases when the most of SAR image pixels in the



computation window has similar brightness values, i.e. the image is homogeneous and contains only a few gray levels. In the frame of SAR image interpretation, the most non-structured areas have higher values, e.g. OW calm or windy OW that has no visual effects (Fig. 5b, i).

Correlation is a measure of the linear relationship between the gray levels of neighboring pixels of radar images. High values (close to 1) indicate a linear dependency between the brightness levels of pixels in the pairs in the computation window and can be obtained for similar gray-level regions. Thus, the any signal variations increase correlation values, representing, e.g. a mixture of various ice types or border of different uniform surfaces (Fig. 5j).

Inertia or contrast characterizes a measure of the local variations presented in an image, i.e. where a huge number of pixel pairs have different brightness values, and shows the difference of brightness. Therefore, if the radar image has more heterogeneous texture character and if there is a large amount of brightness variation, inertia has higher significance that makes objects distinguishable. In the case of visual SAR analysis, a border between heterogeneous sea ice area and open water area, which consist of very close ice floes and//or broken ice, has the highest values (Fig. 5c).

Homogeneity or Inverse Difference Moment determines the uniformity of image and has a high values at the most insignificant change in the brightness levels, i.e., at the lowest contrast of homogeneous images. Entropy measures the randomness of the image texture heterogeneity (intensity distribution), it has a maximum value when all the elements are equal, and characterizes the image ordering. The entropy increases with low variability in the computation area of radar image and indicates a random mixture of scattering mechanisms. High values could indicate ice edges and deformations that create a few strong reflections. Inhomogeneous areas will also produce enhanced values, due to intensity differences in the mixture components, even when the radar reflections are not particularly strong. An inhomogeneous image has lower entropy and homogeneity than a homogeneous image, and in fact, when energy gets higher, entropy is lower (Fig. 5e, k, l).

Cluster prominence is a measure of the skewness of the matrix or, in other words, the lack of symmetry. When cluster prominence is high, the image is not symmetric with respect to it texture values. For the image, this means that there is little variation in gray-scales of SAR image. The third statistical moment is a measure of the distribution asymmetry around the mean of intensities in SAR images and it is named the skewness (Fig. 5d, f) (Unser, 1986; Albregtsen, 2008).

Sea ice in upper part of Fig. 5a could not be distinguished from rough open water (upper right). However, the Fig. 5h shows the reliable detection of sea ice-covered area (left side). The scatterplot on Fig. 4 represents advantage of texture feature application for discrimination between the sea ice and two classes of open water using both polarizations, where sea ice (green) can be clearly seen as standing separately from OW (blue). The same situation is shown on Fig. 5b and i – the dark blue colored ice on the edge near open water (Fig. 5b) is clearly seen, but other ice-covered area can be incorrectly defined as open water, whereas Fig. 5i add more useful information about open water location (blue colored area). The scatterplot on Fig. 4b represents the separation of sea ice and rough open water classes. The scatterplot Fig. 4c demonstrates how different texture characteristics, e.g. homogeneity versus 3rd moment, of different polarizations can add useful information for detection. The examples on Fig. 5c, g, and i show that texture features can be used in applications to obtain well-delineated borders.



### 5.4 Sources of errors

The MET Norway manual products and our algorithm results show generally a good consistency. However, differences typically appear at the ice – water boundary and inside ice-covered areas, where leads or channels on the SAR image are not delineated on the MET Norway ice charts. This indicates that our algorithm is capable of representing more detailed ice – water boundaries. Some differences are also found in the coastal zones, where narrow ice zones near the coast are wrongly shown in our results or fast ice is wrongly classified as OW by our algorithm. The first misclassification can be explained by the low resolution of the land mask derived from MODIS data: the land mask highly underestimates the actual land area after it is rescaled to the RS2 data resolution. The second misclassification can be explained by appearance of fast ice and calm open water on a SAR image and its similarity in the low backscatter. For this case the polarization difference in backscatter between HH and HV bands (cross-polarization ratio) could be included for further improvement (Sandven, 2008; Dierking and Pedersen, 2012; Moen et al., 2013). More significant classification errors can be found in the MIZ. This area is particularly difficult to classify automatically due to its very smooth ice signature.

Detecting typical backscatter ranges and textural structures for different sea ice types and water areas with different roughness stages is extremely difficult due to the high dynamic and variable nature of sea ice and wind speed impact. In particular, different structures on the water affected by wind and visually detected on the SAR images (e.g. stripes, eddies, etc.) may cause wrong sea ice classification.

Residual HV noise effects (after correction) along the ScanSAR image beam boundaries and false signal variations inside the separate beams (Fig. 5 h-l) can have an uncorrected effect on the texture feature analysis and may cause classification errors. These residual noise effects are not visible in the ice-covered areas, but rough open water on high incidence angle close to the beam boundaries may be erroneously classified as sea ice.

The backscatter signal of melting ice becomes similar to open water and imposes limitations for the classification of RS2 images for the summer season.

We assume that our automatic algorithm classifies SAR images more reliable than represented by the provided accuracy (91 %), and this inconsistency may occurs for the following reasons:

1) Different resolution and disagreements in interpretation: The MET Norway ice charts have a lower resolution than our automatic ice charts, making an absolute accurate estimation the ice conditions in the each SAR images and detailed comparison impossible.

2) Different classes: The classes obtained by MET Norway are not consistent with the simple ice water classification provided by the algorithm. In the comparison, we reclassify the MET Norway ice chart into ice and open water using a threshold of 15%. This assumption appears to be the subjective error factor during the validation process and finally reduces the accuracy.

3) Different timing: MET Norway provides manual ice charts for every working day, but not for weekends and holidays. This might cause a difference in timing up to several days. Manual and automatic ice charts of the same day might also not



be based on images taken at the same time of the day. Fram Strait is a very dynamic region and the sea ice situation can significantly change over time periods of several hours.

Considering these limitations of the comparison, we assume that the algorithm performs correct classification in more than 91% of the cases.

## 6 Conclusion

We have proposed an automated OW / ice cover classification of RADARSAT-2 SAR ScanSAR Wide beam mode data, acquired over Fram Strait with different wind speed conditions, which uses backscatter and texture features together in a SVM approach. The higher wind increases the contrast of open water area on different polarizations (HH and HV), that shows bright areas at HH against dark background for HV channel, and the water is distinguished more reliably on dual-
10 polarized RS2 data.

Experiments with ENVISAT ASAR HH data from similar scanning regime (Wide Swath Mode) showed a similar trend of backscatter dependence on incidence angle, and the same technique was applied for HH-band of RS2 SCW images. The ScanSAR image swath consists of different combinations of four physical beams, and there are well-known technical features caused by a wave-like modulation of the image intensity in azimuth direction throughout the entire image in the sub-
15 swaths and their edges of HV band (Romeiser et al, 2010). Although the techniques for compensating the effect in the SAR processor have been developed and applied, some ScanSAR images are still showing residual effects. To improve utilization of such images, we have carried out a procedure of HV band noise reduction that is applied as a pre-processing tool. Computing texture features with sliding window size of $64 \times 64$ pixels and number of backscatter quantized gray levels amounted 32, we obtain the SAR image classification results for period from January, 2013 until October, 2015 and more
than 2700 scenes were processed. The results show that SVM texture algorithm has good discrimination between open water and sea ice areas with accuracy 91% compared with ice charts produced by MET Norway service.

Sentinel- 1A satellite launch have required the new transformation of the algorithm for utilization of the SAR imagery in our sea ice mapping application to implement automated processing chain. The automated SVM based algorithm has been adopted for operational decoding of open water and ice boundary, and it will also be extended and improved for sea ice type
classification (Korosov, 2016).

### Acknowledgements

The work has been supported by the EU projects MyOcean (Grant agreement no. 218812), SIDARUS (Grant agreement no. 262922), MAIRES (Grant agreement no. 263165), Research Council of Norway (contract 196214), Norwegian Space Center (JOP.01.11.2), and EuRuCAS (Grant agreement no. 295068), SONARC project under Research Council of Norway (contract
no 243608) and Russian Foundation for Basic Research (RFBR) project SONARC (Grant agreement no. 15-55-20002). We





acknowledge the MDA for providing RADARSAT-2 data through SIDARUS project. Authors would thank Nick Hughes from Norwegian Meteorological Institute for providing high resolution ice charts for validation of the classification method.

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

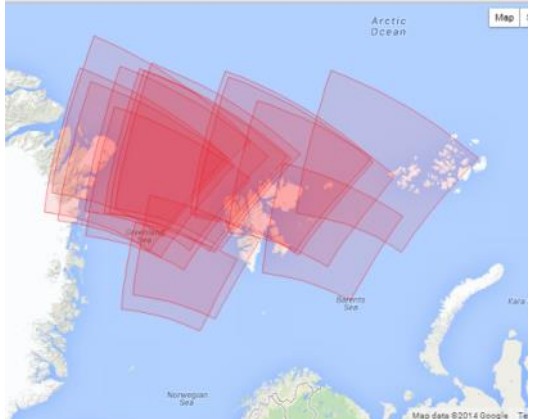

**Figure 1. Location of RADARSAT-2 image used for training. All data are provided in GeoTIFF format with auxiliary XML files by Center for Earth Observation and Digital Earth (CEODE).**



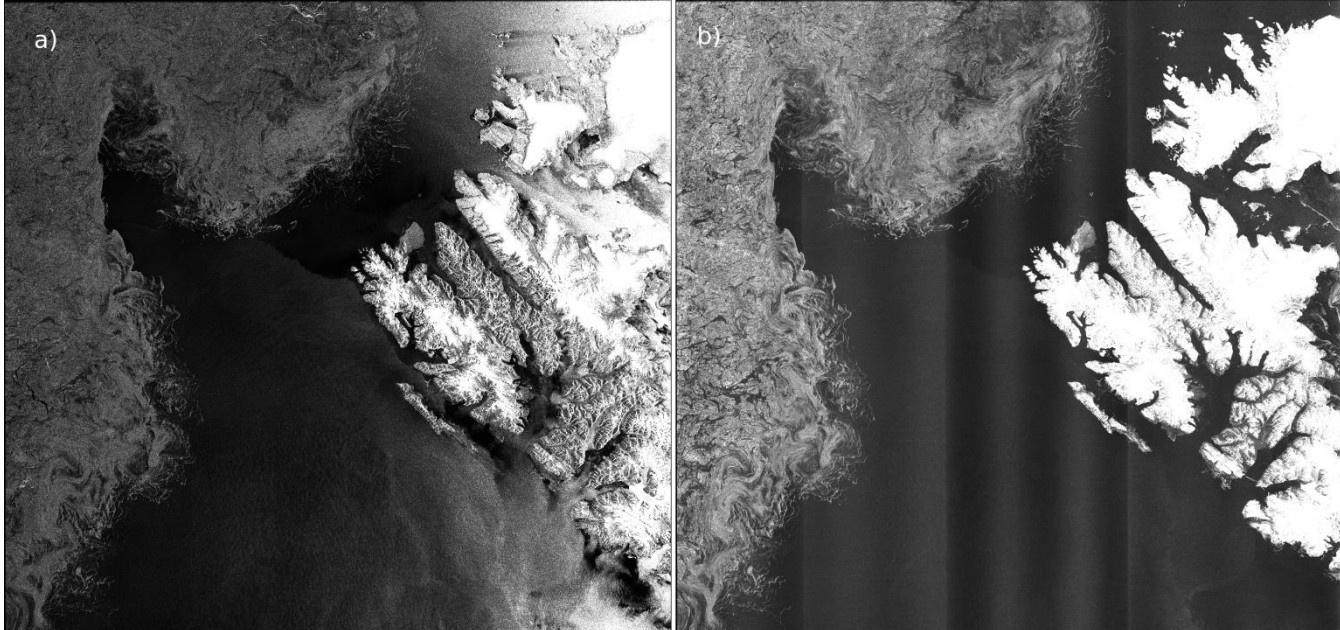

**Figure 2. RS2 SCWA dual-polarization image taken over Fram Strait on November 28, 2011 prior pre-processing. a) HH channel with angular dependence; b) HV channel with along track noise floor variations.**





**Figure 3. RS2 SCWA dual-polarization image taken over Fram Strait on November 28, 2011 including pre-processing. a) Calibrated image after correction of σ° at 35° incidence angle using predefined coefficient for sea ice = 0.298; b) Noise corrected image: beam boundaries are visible due to differences in noise levels between adjacent beams; c) σ° curves of SAR image across the entire swath: original image (blue) and after angular correction (green); d) σ° curves of SAR image along the whole swath. The blue curve shows σ° value profile of the raw HV channel image over the horizontal line, the red curve depicts the noise floor level and the green curve is the result of subtraction.**

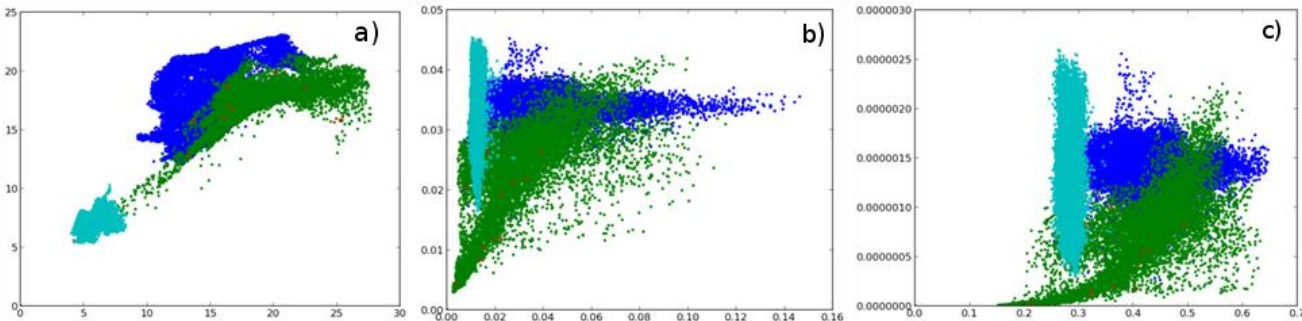

**Figure 4. Scatter plots of texture features combinations. The scatter plots show how a couple of textural features calculated from RS2 images, shown in Fig. 1, can be used to classify ice (green), OW (blue) and calm OW (cyan). a) σ° of HV vs sigma0 of HH; b) Energy of HV vs energy of HH; c) Homogeneity of HV vs 3rd moment of HH.**





**Figure 5. Set of texture features calculated for RS2 SCWA scene, November 28, 2011, the Fram Strait. a) Backscatter, b) Energy, c) Inertia, d) Cluster prominence, e) Entropy, f) 3rd statistical moment of brightness, and g) Standard deviation of HH-polarization. h) Backscatter, i) Energy, j) Correlation, k) Homogeneity, and l) Entropy of HV-polarization. The legend shows the normalized feature values for each plot.**





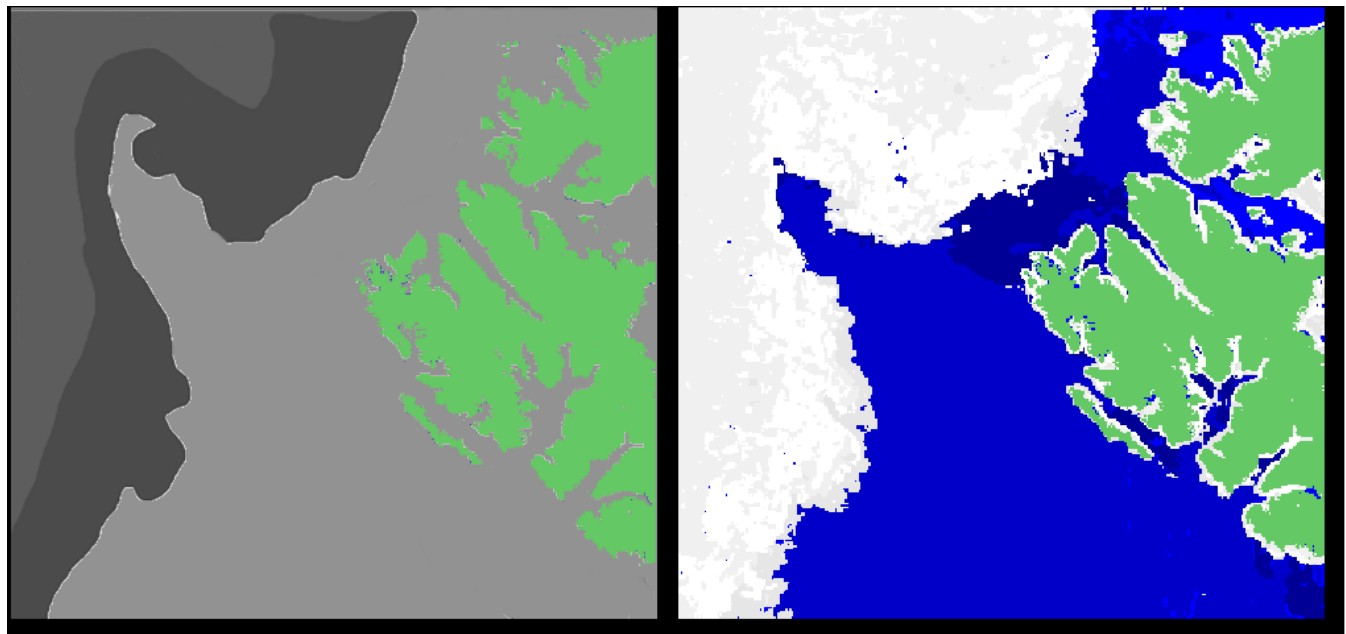

**Figure 6. OW/sea ice classification of RS2 SCWA image shown in Figure 2. a) Manual classification based on sea ice expert analysis to delineate sea ice (in the MIZ and general sea ice cover) and open water (calm and rough open water): dark gray – sea ice; very dark gray – marginal ice zone; light gray - OW; green – land. b) Automatic SVM classification result: white - sea ice; dark blue – calm OW; blue - OW; green - land.**

**Table 1. Monthly averaged accuracies of the automatic ice charts compared to MET Norway ice charts (results given in %)**

| | | 2013 | | | | | | 2014 | | | | | | 2015 | | | |
|---|---|---|---|---|---|---|---|---|---|---|---|---|---|---|---|---|---|
| months | images | ov acc | std | ow err | ice err | months | images | ov acc | std | ow err | ice err | months | images | ov acc | std | ow err | ice err |
| Jan | 72 | 91.52 | 5.43 | 3.99 | 4.50 | Jan | 97 | 91.89 | 4.70 | 2.52 | 5.59 | Jan | 51 | 94.84 | 3.10 | 1.28 | 3.88 |
| Feb | 70 | 91.05 | 4.54 | 2.66 | 6.30 | Feb | 93 | 92.11 | 5.05 | 3.37 | 4.52 | Feb | 33 | 94.47 | 4.05 | 2.33 | 3.86 |
| Mar | 106 | 91.21 | 4.71 | 1.20 | 7.59 | Mar | 110 | 92.20 | 3.45 | 2.83 | 4.98 | Mar | 73 | 94.36 | 4.40 | 1.67 | 3.82 |
| Apr | 110 | 92.03 | 4.57 | 0.95 | 7.02 | Apr | 130 | 93.34 | 3.40 | 1.30 | 5.36 | Apr | 54 | 94.86 | 4.36 | 1.47 | 3.83 |
| May | 111 | 88.60 | 7.96 | 0.88 | 10.52 | May | 137 | 92.80 | 4.77 | 1.00 | 6.20 | May | 63 | 95.05 | 3.21 | 0.72 | 3.81 |
| Jun | 98 | 87.64 | 7.58 | 1.59 | 10.76 | Jun | 93 | 89.98 | 5.78 | 1.54 | 8.48 | Jun | 67 | 84.73 | 14.09 | 0.69 | 3.80 |
| Jul | 83 | 89.73 | 8.01 | 2.72 | 7.54 | Jul | 95 | 86.82 | 9.89 | 1.98 | 11.20 | Jul | 47 | 74.49 | 21.61 | 1.73 | 3.81 |
| Aug | 85 | 94.36 | 3.10 | 2.96 | 2.68 | Aug | 88 | 88.39 | 10.87 | 1.87 | 9.74 | Aug | 47 | 86.65 | 12.25 | 2.64 | 3.85 |
| Sep | 93 | 95.88 | 2.02 | 2.47 | 1.65 | Sep | 97 | 87.55 | 17.56 | 8.24 | 4.21 | Sep | 43 | 94.83 | 3.87 | 3.36 | 3.78 |
| Okt | 72 | 94.53 | 2.99 | 3.98 | 1.49 | Okt | 78 | 94.89 | 3.15 | 1.87 | 3.24 | Okt | 27 | 94.69 | 4.16 | 4.58 | 3.78 |
| Nov | 84 | 92.00 | 4.77 | 5.10 | 2.90 | Nov | 47 | 94.58 | 2.84 | 2.38 | 3.04 | Nov | | | | | |
| Dec | 97 | 90.93 | 6.63 | 3.18 | 5.88 | Dec | 54 | 92.94 | 7.99 | 3.45 | 3.61 | Dec | | | | | |

**ov acc - monthly overall accuracy; std - standard deviation; ice err – sea ice on MET Norway ice chart, OW on automatic ice chart; OW err – OW on MET Norway ice chart, sea ice on automatic ice chart**





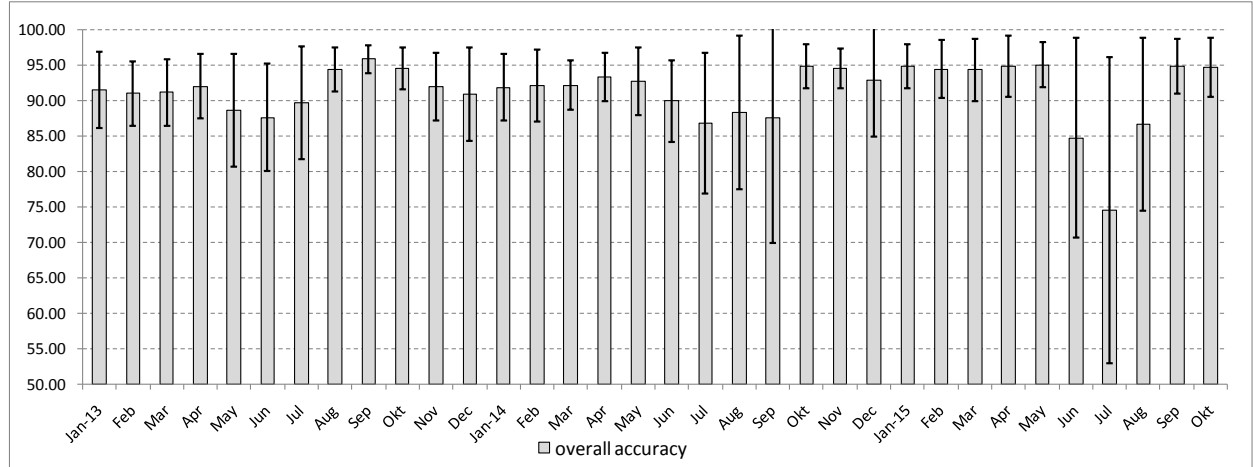

**Figure 7. Monthly accuracy and standard deviation of SVM classification of RS2 images assuming that MET Norway operational ice charts are correct.**





**Figure 8. Validation procedure of automatic classification results compared to MET Norway ice charts. a) Original RS2 SCWA SAR image (HH–polarization), taken over the southern part of Svalbard on March 14, 2013; b) Collocated subset of manual ice concentration chart, provided by the Norwegian Ice Service (met.no) for the same day; c) Result of the SVM classification; d) Result of the SVM classification with delineation of 2 classes: water, sea ice; e) Ice chart of MET Norway reclassified into two classes: open water (ice concentration from 0 to 15%) and sea ice (ice concentration from 15% to 100%); f) Difference of recalculated MET Norway chart and classification result - represents error matrix as "image": no difference, sea ice error (sea ice at MET Norway, OW at our results), OW error (OW at MET Norway, sea ice at our results). Overall accuracy is 95.78%, OW error is 0.19%, and ice error is 4.03%.**