# Peer review of "Operational algorithm for ice/water classification on dual-polarized RADARSAT-2 images"

_The Cryosphere, 2016_

## Referee Comment (RC1) · Anonymous Referee #1 · 14 Jul 2016

Review of "Operational algorithm for ice/water classification on dual-polarized Radarsat-2 images" by Natalia Zakhvatkina and 4 co-authors

General comment: The task of developing robust algorithms for operational sea ice classification is important for obvious reasons. The process of development includes also interesting scientific questions, in most cases related to image processing techniques. Recently, the combination of different polarizations has gained increasing attention because of radar systems capable of acquiring data at two or more polarizations at the same time.

The use of dual-polarization is the main topic of the present manuscript and of topical interest not only with respect to Radarsat-2 but also to ESA's Sentinel-1 mission. I am in particular impressed about the fact that more than 2700 Radarsat-2 images were classified and compared to operational sea ice maps provided by Met. Norge. This enabled the authors, e. g., to judge the classification accuracies under summer conditions. The paper is clearly structured. However, several passages in the text need to be clarified concerning their content, and the English should be corrected. The largest problem I see is the lack of information about the criteria which were applied to decide which combinations of textural features (out of a group of 9) are optimal for separating sea ice and open water (separately for HH- and HV-polarization), and why?

I recommend accepting this paper after major revisions. I will not provide suggestions for improving the language nor will correct write errors but leave this to the (co-)author(s).

Detailed questions:

(1) page 1, line 24: what is meant by "ice edge dynamic distribution" – is it "(dynamic) ice edge variations"?

(2) page 2, lines 1-2: "…based on surface roughness and other characteristics of the scene". Surface roughness is an ice property but not a scene characteristic (like texture). What you probably mean is "…based on gray tone variations (caused by variations of the backscattered radar intensity) and other…"

(3) page 2 lines 4-5: should be rephrased…"…and open water can have similar $\sigma^0$, depending on wind speed and direction…"

(4) page 2, line 7: "…image texture and OTHERS…" – one more example for "others"?

(5) page 2, lines 27-31: (a) It is not true that grey ice and multi-year ice show similar radar brightness at HH-polarization in any case – usually multi-year and first-year (level) ice can be very well distinguished at C-band HH-polarization. Only during summer the differences between young and old ice diminish. (b) When does new ice appear bright? Usually it appears dark! The situations for a bright appearance that come into my mind are frost flowers or broken and deformed new ice (which due to deformation reveals a very rough surface). (c) At HV-polarization smooth thin ice appears dark, and open water appears dark as well, independent of the wind speed. The examples (a) – (c) have to be corrected and/or to be explained more in detail.

(6) page 2, lines 31-34: Does already the first sentence ("The dual polarization…") refer to the work by Geldsetzer and Yackel, 2009, or to another study?

(7) The goal of the present study is well described on page 3, lines 20-23. However, in my opinion it would be very useful if the authors emphasize the special motivation of their work, considering the fact that studies on classification with dual-polarization data were already published earlier.

(8) page 4, lines 6-8: This is an important point of your work! You should emphasize here in addition that you have to analyze a larger number of radar images because the radar intensity contrast between open water and ice is varying significantly dependent on ice conditions and wind speed/direction (the latter affecting the radar brightness of open water).

(9) page 4, lines 13-14: Sentence: "The reason is…" I recommend discarding this sentence. Actually, the magnitude of the received cross-polarized signal depends on the structure of the illuminated target. In case of sea ice, e. g., the cross-polarized return increases with increasing macroscopic ice deformation.

(10) page 4, lines 14-16: "This causes…" Does the low intensity at HV-polarization explain its variation in across-track direction? I don't think so. I guess the reason is technical limitations of the radar sensor electronics when operated close to their noise level. Please check the Radarsat-2 manual. A good description of the HV-variations can also be found in Komarov & Barber, TGRS, Vol. 52, No. 1, pp. 121-136, 2014

(11) page 4, line 22: start "Our AUTOMATED…" Your automated algorithm does not include 6 steps but only 3 as you yourself explain on top of page 5. What you describe following this sentence is the first step of your analysis, namely the (manual) determination of thresholds and suitable textural measures that are later used in the fully automated (unsupervised) classification. Please formally separate the training/test part and the subsequent automatic classification more clearly.

(12) page 4, line 29-30: Do you mean "Training of the automatic classifier (e. g. SVM) using different combinations of texture features together with radar intensity, based on manually classified images"?
(is it really a "training" or rather a determination of thresholds between ice and water?)

(13) page 5, line 5: how large is the sliding window? (You can give a hint to section 3.4 in which you mention the sizes).

(14) section 3.1: I recommend that you provide the equation used for the incidence angle corrections and define the "linear-trend coefficient" that you use later. You should also give a hint that you discuss problems with this correction in section 5.1.

(15) page 5, line 24: "incontinuity" = discontinuity? Since the problematic zones are masked out in the HV-images, one should also see them in the figures that you present later. But masks are not included in the figures.

(16) page 6, sentence lines 2-4: I assume that all co-authors understood this sentence but I do not. "…trained classifier…" = ice analyst from Met. Norge? If the images show, e. g., cracks, ridges and leads, why can't the "trained classifier" not identify them (they are distinct features)?

(17) page 6, lines 4-6: please separate the single sub-classes more clearly. Is it 1: young ice, first-year ice and multi-year ice, 2: fast ice, 3 broken ice mixed with ice-free water? What is the reason to group the ice types like that? Usually, e. g., young ice and multi-year ice reveal very different radar signatures (at least in intensity, not necessarily in GLCM textures). Younger ice can be mis-classified as open water at lower wind speeds, multi-year ice as open water at higher wind speeds, if one focuses on radar brightness.

(18) page 6 line 14: "the full range" refers to the number of grey tones, which are rescaled to a lower number of bits? How many bits do your original images have?

(19) page 7, lines 5-8:  this is another very important part of your work which needs to be described a little more in detail. (a) You vary the computational parameters: window sizes, co-occurrence distances, and quantized grey levels (please mention the increments for the latter two). With these different computational parameters you calculate the different texture features - which means that you have a huge number of possibilities. By which method did you determine the optimal combination of computational parameters (which you mention on page 8 lines 25-30)? Is it

described in your TGRS paper from 2013, to which you refer to in section 4.2? The TGRS paper, however, does not cover all aspects of the processing described here. (b) Which criteria did you apply for deciding which combination of texture features is optimal for classification at HH- and at HV-polarization? (In your TGRS-paper you used all texture features.) (c) How many radar images and related ice charts from Met. Norge did you use for these tests? (d) Were the selected combinations of texture features best for all images, or only in a majority of the investigated cases?

(20) Section 3.6: For ice chart production, also optical images were employed. In how many cases could they not be used because of dense cloud cover? Did you use a "weight" indicating the reliability of the ice charts (assuming that the lack of optical information causes more difficulties for the ice analyst to separate ice and water)?

(21) Page 8, line 15: what do you mean by "…and increases from top right to bottom left"? Do you refer to the extent of the open water area?

(22) Page 8, lines 17-18: Does new and thin ice really always appear brighter than OW at HV-polarization? I doubt this.

(23) Section 4.1: As already noted above (comment 15): How did you handle the computation of texture features in the zones along the borders between the image stripes that you indicated as white lines in 3b?

(24) Section 4.2: Please see comments (19). From my point of view it is up to you whether you provide more details about methodological aspects in section 3.4 or in this section 4.2.

(25) Page 9, lines 3-5, Fig. 4: The usefulness of the different texture features is not clear to me. When combining the intensities at HH and HV, calm OW is very well separated, but OW and ice partly overlap. The latter is also valid for the two graphs to the right, but here even calm OW overlaps wit the ice. Please provide some convincing arguments why the texture features improve the water-ice separation? (See also comment 33 below).

(26) Referring to Fig. 6, 7, 8: why is Fig. 5 not mentioned before them?

(27) Section 4.4: How was the comparison between classified radar image and Met. Norge ice charts carried out? Were the latter digitized to the pixel size of your radar images?

(28) page 9 line 28-30: (a) "…ice charts WERE obtained manually…"; (b) what do you mean by "higher rate of thematic processing" in this context?
I think that also ice charts from Met.Norway could include a higher level of details if the ice analyst delimits small-scale features. But this would require a longer time for ice chart production. What is the processing (computer) time needed when applying your algorithm compared to the semi-automated chart production by Met. Norge? This aspect is important and should be added to the discussion section 5.

(29) Section 5.1: (a) The title refers only to the incidence angle correction, but the noise correction is also dealt with, at least in the last sentence.

(30) page 10, lines 5-6: "Strong dependence of the backscatter on incidence angle in open water surface… is significantly higher than for sea ice". This statement is not true in any case, since the incidence angle sensitivity depends on the surface roughness: the smoother the surface, the larger the sensitivity (in particular close to the angle of specular reflection). Thin level ice can be very smooth and may show a larger sensitivity than a wind-roughened water surface.

(31) (a) page 10, line 8: what do you mean by "overestimated signatures"? (b) Line 9: what do you mean by "ice cover has more reliable backscatter ranges for various ice types"? Since wind speed

can be determined from radar measurements over open water, this means that also open water has clear defined ranges of backscattering intensity for a given wind speed and direction (except for OW patches within the ice cover and at the ice margin where ocean surface wave interactions are more complicated).

(32) Section 5.2: (a) You should make clear that in the first sentence you refer to the nine texture parameters given by equations 2-10. If you in addition use intensity and standard deviation, there are in total 22 parameters considering both HH and HV (and not 20). In the following text you should make clear that you now refer to section 4.2, in which you selected 5 textural parameters + intensity + standard deviation for HH, and 4 textural parameters + intensity for HV, leaving in total 12 parameters. (b) I do not understand the meaning of the two sentences from line 20 to line 23. E. g., what are "poor features"? What is a discriminant function in this context, and why is it needed? If all features are used, does this mean that the classification accuracy is lower in any case? There is of course an optimal number and an optimal selection of textural features giving highest classification accuracy, but with, e. g., only 4 textural parameters you might theoretically still get better accuracies than with less parameters. The last 3 sentences of section 5.2 refer to the reduction of dimensionality but a direct link to the own processing described in section 4.2 is missing (e. g. the criteria for selecting some of the nine given textural features and excluding others).

(33) Section 5.3: With this section I had considerable difficulties. (a) I suggest that you provide a table giving the function of each textural parameter (e. g. measure of local variations), and the interpretation of the respective parameter related to Fig 5.
(b) In this section you introduce alternative denotations for the textural parameters (e. g. energy – angular second moment; homogeneity – inverse difference moment etc). You should make this clear by expressing it like "Energy (also called 'Angular Second Moment')…" (after this, it is fine to go on with e. g. "Homogeneity or Inverse Difference Moment".
(c) page 10 line 30. I found that the "Energy" is the square root of the Angular Second Moment. Is "repeatability" another denotation for energy? In Fig, 5i, the OW area, which appears homogenous in Fig. 3b, reveals a very low energy – is it a noise effect? The bright blue zones in the OW-area in Fig. 5i are due to the stripes in Fig 3b?
(d) page 11 sentence lines 6-7: I do not understand this sentence. The GLCM correlation function is calculated for a co-occurrence distance of 8 pixel? Fig. 5j: I do not understand why the correlation is very high in the marginal ice zone and low in the more closed ice. I would expect it vice versa. The low correlation value over open water is again a noise effect?
(e) page 11, line 13: homogeneity – why is the homogeneity high in the marginal ice zone and low in the inner ice zone and over water (Fig. 5k)? Again I expected this vice versa.
(f) page 11, line 17: "…indicates a random mixture of scattering mechanism". I think here the entropy from the entropy-alpha decomposition described by Cloude et al. (where entropy is indeed related to the character of the scattering mechanism) is mixed with the GLCM entropy, which does not give any information about the scattering characteristics from within one pixel but relations between neighboring pixels.
(g) page 11, lines 26-28 and lines 30-33, regarding Fig. 4: I do not see a clear separation between OW (dark blue) and sea ice (green), and the separation capability of the texture features seems to be worse than the one of the intensities. See also comment 25 above.

(34) Section 5.4, first paragraph: (a) it should be considered that the maps drawn by the ice analysts are a "smoothed version" of the ice cover variations. In principle the ice analysts could also provide more detailed maps. However, I anticipate that an automated algorithm can do this much faster – see also comment 28 above. (b) why didn't you use a better land mask? (c) page 12 lines 17-20: how did you treat the beam boundaries when calculating the texture features (see also comment 23 above)? (d) page 12, lines 32-33: why didn't you exclude cases in which the temporal difference between manual and automatic ice chart was large?

---

## Referee Comment (RC2) · Anonymous Referee #2 · 22 Jul 2016

**Review of the paper "Operational algorithm for ice/water classification on dual-polarized RADARSAT-2 images" by N. Zakhvatkina and 4 co-authors**

**General comment**

Currently not many SAR based products are in operational use in the sea ice monitoring although a large amount of case studies utilizing just a few SAR scenes have been carried out. In the present paper one such operational product is introduced and tested. The proposed approach produces an ice/open water chart in the Arctic and it can be used around the year. It uses C -band dual-polarized SAR imagery acquired by the SAR sensor in the RADARSAT-2 satellite. The approach is applicable also for Sentinel-1 data.

The major strength of the present paper is that the presented classification scheme is applied to over 2700 RS2 SAR scenes covering a time period of almost three years from 2013 to 2015. The classification results are validated against manually derived ice charts provided by the Norwegian Meteorological Institute. According to the validation procedure the accuracy of the results is high (about 91 %). Even during the summer months the accuracy remains high which is a remarkable achievement. It is well-known that ice conditions during the summer are difficult to analyze using SAR data. The article follows the traditional structure of a scientific research paper. However, the language needs editing.

There are several issues that the authors must address before the publication. I think that the main problem of the paper is that the authors often support their claims with words instead of calculations. The texture features have a central role in the classification. Despite this the authors have not demonstrated their importance quantitatively. For a reader it remains unclear how much the addition of these features increases the classification accuracy. I am more specific in my detailed comments.

When I wrote my review I had an access to the review of the Referee 1. This helped me a lot because my colleague raised several important points and questions which were also in my mind. I have focused in my comments mostly on questions not discussed in the first review.

I recommend accepting this paper after major revisions.

**Detailed comments**

1. P2L26- Dual-polarization has several ..

   C: The major advantage offered by the HH and HV polarizations is that they are results from different backscattering mechanisms. $\sigma^o_{HH}$ is dominated by first-order scattering (direct backscatter with no multiple reflections), whereas $\sigma^o_{HV}$ is a result of multiple scattering (two or more reflections involving two or more scatterers)). Hence it easy to understand that the magnitude of $\sigma^o_{HV}$ is usually smaller than the magnitude $\sigma^o_{HH}$. The energy radiated towards the radar decreases significantly with each reflection. Rewrite also the text in P4 L11-14 keeping in mind the above explanation and the comment (9) by Referee 1. E.g. increased ice deformation increases also the amount of the multiple scattering as does the large volume scattering component from MYI.

2. P3L20-23. C: Here the authors could also comment why they have not targeted to produce a sea ice concentration chart which would provide to the users and modelers more information than a binary open water/sea ice chart. As we can see from Fig. 8 the presented classification chart (8d) is not a good approximation for an ice concentration chart (8b).

3. P4L5-8. C: Here you could add a remark that the classifier trained in the winter conditions is not ideal for the summer conditions.

4. P5 Sect. 3.1. and P8 Sect. 4.1. C: I strongly support the suggestion (14) of Referee 1. Otherwise the statistical incidence angle compensation that you use is left unclear. In Sect. 4.1 you give just number (0.298) without units and with wrong sign. I assume that you mean the slope coefficient $-0.298dB/1^o$. I wonder why the magnitude of the coefficient is much larger than $-0.196dB/1^o$ given in your 2013 paper for MYI or the coefficient $-0.23dB/1^o$ estimated in Mäkynen et al. (2002) for FYI. It should

be noted that in Mäkynen et al. (2002) the same targets in different images with different incidence angles are examined. If I have understood correctly, in your 2013 paper you have studied targets which looked similar but appeared in different incidence angle ranges in the same image. What kind of procedure have you followed here? Is the steeper slope due only to a different sensor or do some geophysical factors contribute, like sometimes less than 100 % ice concentration in the test images?

5. P6L2-7. C: The referee 1 already commented this passage in the comments (16) and (17) which comments I support. My additional question is that what is the role of these subclasses in the classification scheme. Does SVM use them? If so, how have you selected all these subclasses as input to the classifier. Clarify the text please.

6. P7L5-8. C: An addition to the comment (19) of the Referee 1. Yu et al. 2012 (in your reference list) have applied to the feature selection problem "a forward feature search" which is identical to the forward stepwise selection in the regression analysis. The only difference is that in the feature selection the criterion is the classification accuracy instead of the criteria like AIC, BIC and many others used in the selection of the variables in the regression model.

7. P7: Section 3.5. The description of the SVM is given in a very general level and the text is not well organized. The presentation should be more informative. You have many alternatives to detail your presentation. One is that you formulate the SVM as a solution to an optimization problem (e.g. Hastie et al, The Elements of Statistical Learning, available as a PDF file in the internet) and comment its properties from this point of view. Another approach is to treat the problem as Yu et al. 2012 (mentioned above) have done. In any case you must estimate some parameters when fitting the SVM in your data. Give the estimation method. When someone reads your text, he/she should get an idea what the SVM is and why you have chosen it. The equations are in this context necessary. The SVM gives only a binary classification result. Explain how you have generalized it into the case of three classes (like in Fig. 8c).

8. P8L11. C: Is the radar look direction in Fig. from right to left?

9. P9L3-5. C: I agree with Referee 1 (comment (25)) that Fig. 4 shows no increased discrimination ability with the texture features when compared to the $(\sigma^o_{HH}, \sigma^o_{HV})$ pair. When looking at Fig. 5 my subjective opinion is that Figs. 5a and 5h (corresponding to the HH and HV channels) provide the two best features. Show how the classification accuracy improves when you add texture features to the $(\sigma^o_{HH}, \sigma^o_{HV})$ pair. The sentence in P9L1-2 is not an argument.

10. P9L2. ... methodology description... C: What did you mean by this? In the 2013 paper you selected all the features. Please clarify.

11. Sect. 4.2. and Fig.4. C: How have you normalized the features? As Fig. 4 shows the ranges of different textural features are highly variable. It would also be better if the normalized values of the textures (as in Fig. 5) would be used in Fig. 4. In any case the SVM requires that normalized feature values are used or the distance concept in the radial basis function is arbitrary.

12. Fig. 8. In the figure caption: ...open water (ice concentration from 0 to 15%)... C: How have you identified such areas? The manual ice charts has the ice concentration classes: 0/10 -1/10, 1/10 -4/10 and so on. The class 0/10 -1.5/10 is missing.

13. P9L28-30. C: I disagree with your conclusion that the SVM classification gives a more detailed ice cover map than the manual ice chart. If we inspect Fig. 8b we see how the sea area is divided into subareas with different ice concentrations. In Fig. 8e you have thrown away all this valuable information and forced the manual ice chart to a binary map. The comparison between the automated and manual chart that you have presented in the text is not fair. Please modify your text and assessment.

14. Sect. 5.2. C: An addition to the comment (32) by the Referee 1. Do yo have considered the principal components as a way to deal with the intercorrelation of the features and simultaneously reduce the dimensionality? If you have, why did you reject the principal component analysis.

15. Sect. 5.3. C: As the Referee 1 (the comment (33)) I struggled and often failed to understand your interpretations of the texture measures. This section had to be rewritten, e.g. following the guidelines given by the Referee 1. Just one addition. As far as I know, the only scattering mechanism one is able to measure from the dual-polarized HH +HV image is the depolarization ratio. In the decibel scale the depolarization ratio is simply the difference $\sigma^o_{HV}$ -$\sigma^o_{HH}$.

16. P12L4-5. C: I disagree with you due to the same reason as earlier. I think that for the models a sea ice concentration estimate at coarser resolution is a better option than knowledge of locations of small open water patches or leads.

17. P12L6-8. C: It is possible to derive a land mask from the MODIS data at resolution of 250 m. So the difference is not big compared to RS2 data, especially when we take into account that the resolution of the final product is 1.6 km (P5L6). Why do a MODIS based land mask underestimate the land area? I would expect that it might slightly overestimate it.

18. P12L15. ...different structures on the water affected by wind and .. C: Eddies are not caused by winds. They are results of ocean currents. Write ...affected by wind and currents ..

---

## Referee Comment (RC3) · Anonymous Referee #3 · 25 Jul 2016

Operational algorithm for ice/water classification on dual-polarized RADARSAT-2 images by Zakhvatkina and others.

General Comments: 1. Too often are new image classification algorithms just assessed on a few test images and as a result are not very robust. A strength of this paper is that the algorithm the authors present is tested over a great number of images. In this regard, the classification results are very good over the annual cycle and while their Table nicely summarizes this result it would be useful to showcase the classification results in more detail. Specifically, I think readers would like to actually see (visually) the performance of this algorithm during the summer melt or more difficult classifications – I know, I did after reading this paper. I suggest adding a few more examples or even a panel figure of classification comparisons with ice charts highlighting algorithm performance visually. They do not have to be perfect but for operations that does not

matter – ice analysts want to see how the algorithm will perform in the most difficult conditions.

2. The English structure requires some serious attention. There are numerous passages that are difficult to follow or just do not make sense. I suggest a thorough English edit be required before publication.

3. The manuscript structure can be improved by combining the results (Section 4) and discussion (Section 5) sections. As it reads now, certain sub-sections of the discussion do not reference material created in the analysis which they should do (i.e. 5.1, 5.2). For example, the discussion on incidence angle has no reference to the correction the authors applied. Validation and Sources of Error can be easily compared and would make for a better read.

4. Perhaps more important than 2 and 3, it is not clear from the text (4.2/5.2) how the optimal texture combinations where chosen? I think this needs to be addressed in the methodology not the results. Nevertheless, this a remains a major problem that needs to be clarified.

Overall, I think the algorithm presented in this paper is worthy of publication when the comments outlined above and below are taken into consideration.

Specific Comments/Questions P1L22: I don't think exploring is the correct word. Perhaps quantifying?

P1L24: regions not region

P1L25: such as ERS-1/2

P1:30: that extend operational utility.

P2L1: The objective of sea ice classification is to identify sea types and open water. You do not need "based on" unless you are going to mention everything taken into consideration.

P2L5: That is basically all factors. Why not just say discriminating between open water that is wind roughened and sea ice is difficult? P3L4: The CIS did not developed MAGIC. To my knowledge it was developed by Dr. Clausi at the University of Waterloo.

P3L20: The goal is not to extend ENVISAT single polarization, it is simply to utilize dual polarization data for ice classification.

P3L30: ice conditions.

P4L10-20: No need to describe what HV is. Start with: The HV channel...but this is a difficult passage to follow on the physics as to why HV is darker than HH in RADARSAT-2. I suspect English is the root cause. Revise.

P4L22-33: It would be better if the methodology was written out in paragraph form rather than numbered points. You can still include numbers (i.e. i, ii, iii, etc) in the text.

P5L13: Why not just simply state that the imagery was normalized to 35 degrees and move on? IA correction does not require a separate sub-heading.

P5L30: Unclear what is meant by Manual classification has be done...? Did the author's manually classify the imagery? It is unclear what is trying to be communicate in this sub-heading.

P6L10: Are all the texture features used in the classification or just some? How are certain ones selected over others? This needs to be clear in the text. See General Comment 4.

P8L5: Is there a website link to the MET ice charts? Do they use RADARSAT-2 imagery? If they do not, this should be mentioned as they are an independent source for comparison.

P8:10: Why is figure 6 being introduced before the other figures?

P13L6: developed, not proposed.

Figure 2: Needs some latitude/longitude information for reference

Figure 3: Needs geography similar to Figure 2. The line graphs need to include axis labels and the font needs to be bigger.

Figure 6. Labels a) and b) are not included on the image.
* * *

---

## Author Comment (AC1) · 28 Sep 2016

**RESPONSE TO REVIEWER COMMENTS**

**Reviewer 1:**

**(1) page 1, line 24:** what is meant by "ice edge dynamic distribution" – is it "(dynamic) ice edge variations"?
We agree with this remark and corrected this sentence.

**(2) page 2, lines 1-2:** "…based on surface roughness and other characteristics of the scene". Surface roughness is an ice property but not a scene characteristic (like texture). What you probably mean is "…based on gray tone variations (caused by variations of the backscattered radar intensity) and other…"
We agree with this remark and corrected this sentence.

**(3) page 2 lines 4-5:** should be rephrased…"…and open water can have similar $\sigma_0$, depending on wind speed and direction…"
We corrected this sentence.

(4) page 2, line 7: "…image texture and OTHERS…" – one more example for "others"?
We agree with this remark and corrected this sentence.

**(5) page 2, lines 27-31: (a)** It is not true that grey ice and multi-year ice show similar radar brightness at HH-polarization in any case – usually multi-year and first-year (level) ice can be very well distinguished at C-band HH-polarization. Only during summer the differences between young and old ice diminish.
**(b)** When does new ice appear bright? Usually it appears dark! The situations for a bright appearance that come into my mind are frost flowers or broken and deformed new ice (which due to deformation reveals a very rough surface).
**(c)** At HV-polarization smooth thin ice appears dark, and open water appears dark as well, independent of the wind speed. The examples (a) – (c) have to be corrected and/or to be explained more in detail.

(a) We meant the young ice (gray and gray-white): young ice and MYI at HH-pol are rather similar, and, of cause, they both are very well distinguished against level FYI. Young ice and multiyear ice show near similar brightness in the HH channel as the most similar in visual interpretation, we mentioned that their NRCS ranges are overlapped, but not the same.
We can illustrate what we mean: we have done some analysis for ENVISAT and RS2 data. For RS2 is not so satisfactory due to a few SAR data being in use for this analysis. The plots show the different sea ice types backscatter values, averaged in a 10 x 10 square samples, obtained from SAR images of ENVISAT (WSM, HH-pol) and RS2 (SCW, HH-pol).

[Figure]

Envisat ASAR WSM data(winter months, 2011), HH-pol, angular corrected at 31 deg: a) min and maximum of backscatter values, b) mean and STD of backscatter values

5    RS2 sigma0 distribution, winter 2012,2013, HH-pol, angular corrected at 35 deg.

We used term "young ice" following the WMO Nomenclature, and it includes grey (10-15 cm) and grey-white ice (15-30 cm). We understand that high backscatter of young ice (mostly grey ice) in the most (but not all) cases is due to complete coverage of its surface with frost flowers. According to
10   Onstott (1992), the formation of frost flowers act to wick up brine and then form roughness elements on ice surface. An increase in surface roughness translates into an increase in backscatter. In some cases, high backscatter of young ice can be caused by its rafted/ridged to 100% surface. Wind-roughened open water also has high backscatter, which exceed that of young ice in many cases.

15   (b) Yes, we understand that new ice usually appears dark at HH and HV, and the sentence was written incorrectly. Affected by wind and surface currents the new ice sometimes forms a dense cluster, which has appearance the form of typical dark stripes and spots on the SAR images. These stripes and spots as the new ice (including *frazil ice, grease ice, slush* and *shuga*) has darker tone then rough OW, but in some case not so dark as level FYI \ fast ice or OW calm. We kept in mind the newer [younger] ice,
20   which with MYI is difficult to distinguish from sea ice (on different stages of development).

Our previous results showed that the NRCS values of grease ice obtained on ENVISAT ASAR images (HH-polarization, the 23° incidence angle) had range from -17.5 dB to -12.0 dB. In several other cases the NRCS values of grease ice amounted -7.43 dB and -6.15 dB at the start stage of its formation, while the NRCS values of the surrounding rough open water was about -4.0 dB. According to the study [Winebrenner D. P.,. Holt, E. D. Nelson. Observation of Autumn Freeze-up in the Beaufort and Chukchi Seas Using the ERS-1 Synthetic Aperture Radar , 1996] the backscatter of new ice can varied over the wide range - from -23 to -5 dB. Higher NRCS values can be explained by the possible presence of layered nilas, which backscatter is slightly higher than the smooth nilas [Atkinson P. M., A. R. L. Tatnall. Neural Networks in Remote Sensing, 1997].

(c) We agree that at HV-polarization smooth thin ice appears dark, and open water appears dark as well, independent of the wind speed. Here we mentioned that rough OW at HH-pol is rather bright since at HV-pol it is darker, that improves the OW distinguishing from sea ice.

We agree that these statements have some confusion information and corrected the text.

**(6) page 2, lines 31-34**: Does already the first sentence ("The dual polarization…") refer to the work by Geldsetzer and Yackel, 2009, or to another study?
[Geldsetzer and Yackel, 2009] use dual co-polarized C-band SAR imagery for discriminating sea ice types and open water during winter. The analysis was based on ENVISAT ASAR alternating vertical and horizontal polarization (VV, HH) medium-resolution imagery (from their paper).

**(7)** The goal of the present study is well described on page 3, lines 20-23. However, in my opinion it would be very useful if the authors emphasize the special motivation of their work, considering the fact that studies on classification with dual-polarization data were already published earlier.
We have included clarification of our motivation to the paper.

**(8) page 4, lines 6-8:** This is an important point of your work! You should emphasize here in addition that you have to analyze a larger number of radar images because the radar intensity contrast between open water and ice is varying significantly dependent on ice conditions and wind speed/direction (the latter affecting the radar brightness of open water).
We have included clarification to the paper: The radar images include the most typical samples since the radar intensity contrast between open water and ice varies greatly with ice conditions and wind speed/direction which significantly affect the radar brightness of open water.

**(9) page 4, lines 13-14**: Sentence: "The reason is…" I recommend discarding this sentence. Actually, the magnitude of the received cross-polarized signal depends on the structure of the illuminated target. In case of sea ice, e. g., the cross-polarized return increases with increasing macroscopic ice deformation.
We have discarded the sentence, since it was confusing.

**(10) page 4, lines 14-16**: "This causes…" Does the low intensity at HV-polarization explain its variation in across-track direction? I don't think so. I guess the reason is technical limitations of the radar sensor electronics when

operated close to their noise level. Please check the Radarsat-2 manual. A good description of the HV-variations can also be found in Komarov & Barber, TGRS, Vol. 52, No. 1, pp. 121-136, 2014.

The next sentence was rewritten as follows:  The HV channel includes disturbances in azimuth direction (visible as bright and dark stripes) along the burst boundaries in the ScanSAR Wide Beam  SAR image (Fig. 2b).

**(11) page 4, line 22**: start "Our AUTOMATED…" Your automated algorithm does not include 6 steps but only 3 as you yourself explain on top of page 5. What you describe following this sentence is the first step of your analysis, namely the (manual) determination of thresholds and suitable textural measures that are later used in the fully automated (unsupervised) classification. Please formally separate the training/test part and the subsequent automatic classification more clearly.

We agree that the word 'automated' was not unambiguously describing the nature of the algorithm. Nevertheless, the algorithms is from the family of 'supervised classification algorithms' which are commonly understood to include both manual and automatic steps [e.g. Mohri, 2012]. We, therefore, believe that our algorithm also include all 6 steps - 2 of them are manual steps and are performed only one (2 - manual classification, 4 - training of the classifier), other steps are fully automatic. The word 'automated' was replaced with 'semiautomatic'.

**(12) page 4, line 29-30:** Do you mean "Training of the automatic classifier (e. g. SVM) using different combinations of texture features together with radar intensity, based on manually classified images"? (is it really a "training" or rather a determination of thresholds between ice and water?)

It is actually a 'training' (also called 'machine learning'). At the training step (4) a 'teacher' provides values of texture features manually grouped in classes (e.g. ice, water, etc) to the SVM algorithm [BURGES, 1998]. This algorithm creates a hyperplane for categorizing values of TF. These hyperplanes (defined as polynomial coefficient values) are saved after training and are applied at the step (5).

**(13) page 5, line 5**: how large is the sliding window? (You can give a hint to section 3.4 in which you mention the sizes).

We have added: The image size …. window with 16 pixels step size (the detailed parameters are described in Sec. 4.2).

**(14) section 3.1**: I recommend that you provide the equation used for the incidence angle corrections and define the "linear-trend coefficient" that you use later. You should also give a hint that you discuss problems with this correction in section 5.1.

We included the equation with some description and hint.

**(15) page 5, line 24:** "incontinuity" = discontinuity? Since the problematic zones are masked out in the HV-images, one should also see them in the figures that you present later. But masks are not included in the figures.

The masks were excluded from the result images – this is a part of the algorithm.

**(16) page 6, sentence lines 2-4**: I assume that all co-authors understood this sentence but I do not. "…trained classifier…" = ice analyst from Met. Norge? If the images show, e. g., cracks, ridges and leads, why can't the "trained classifier" not identify them (they are distinct features)?

5 We slightly corrected this paragraph: ….. The images selected for our algorithm training did not contain homogeneous ice cover because the mixing of different ice types with different degrees of deformation, cracks, ridges and leads usually occur in ice covered areas.

**(17) page 6, lines 4-6**: please separate the single sub-classes more clearly. Is it 1: young ice, firstyear ice and multi-year ice, 2: fast ice, 3 broken ice mixed with ice-free water? What is the reason to group the ice types like
10 that? Usually, e. g., young ice and multi-year ice reveal very different radar signatures (at least in intensity, not necessarily in GLCM textures). Younger ice can be misclassified as open water at lower wind speeds, multi-year ice as open water at higher wind speeds, if one focuses on radar brightness.

These subclasses were chosen empirically after several algorithm training attempts. We did not only define a number of texture features but also varied the combinations of subclasses. One subclass
15 includes young ice, first-year ice and multiyear ice. The second is fast ice since backscatter and texture has similarity with calm open water. Young ice and multi-year ice can be misclassified as open water at lower or higher wind speeds, but the usage of texture features solved this problem in our case. The main problem was to detect correctly the transition zone from a general ice massif to ice-free area , where can close and very-close broken ice mix with ice-free water. And finally the sea ice class was subdivided
20 into 3 subclasses. In other words, this approach is suitable in the case of obtaining only ice \ water separation.

**(18) page 6 line 14**: "the full range" refers to the number of grey tones, which are rescaled to a lower number of bits? How many bits do your original images have?
25 Originally the RS2 image has 16 bit unsigned integer type and the calibration in Nansat produces sigma0 values in 32 bits floating point format.

**(19) page 7, lines 5-8**: this is another very important part of your work which needs to be described a little more in detail. **(a)** You vary the computational parameters: window sizes, cooccurrence distances, and quantized grey
30 levels (please mention the increments for the latter two). With these different computational parameters you calculate the different texture features – which means that you have a huge number of possibilities. By which method did you determine the optimal combination of computational parameters (which you mention on page 8 lines 25-30)? Is it described in your TGRS paper from 2013, to which you refer to in section 4.2? The TGRS paper, however, does not cover all aspects of the processing described here.
35 **(b)** Which criteria did you apply for deciding which combination of texture features is optimal for classification at HH- and at HV-polarization? (In your TGRS-paper you used all texture features.)
**(c)** How many radar images and related ice charts from Met. Norge did you use for these tests?
**(d)** Were the selected combinations of texture features best for all images, or only in a majority of the investigated cases?

(a) The list of experiments with values of the parameters is now added to the manuscript. The main approach to select the best combination turn out to be empirical testing on several images and qualitative assessment of the results at each step of the algorithm.

We clarified the text in Sections 3.4 and 4.2, and provided one more figure

(b) In TGRS-paper the correlation analysis and visual interpretation of normalized texture features distribution were applied. In this case we use the same technique.

(c) We used 24 training images noted in Sect. 2.

(d) The texture features combinations were the best for the majority of processed images.

(20) Section 3.6: For ice chart production, also optical images were employed. In how many cases could they not be used because of dense cloud cover? Did you use a "weight" indicating the reliability of the ice charts (assuming that the lack of optical information causes more difficulties for the ice analyst to separate ice and water)?

No, we did not use "weight" indicator - confidence level flag information.

(21) Page 8, line 15: what do you mean by "...and increases from top right to bottom left"? Do you refer to the extent of the open water area?

The sentence has been rewritten as follows: The open water area is located on the right-hand side of the image and the ice covered area - in the upper-left corner.

(22) Page 8, lines 17-18: Does new and thin ice really always appear brighter than OW at HV-polarization? I doubt this.

Not always, we mentioned the ice situation on this RS2 image. We agree with this remark and have corrected: The ice-covered areas and the rough OW areas appear both bright in HH and are therefore difficult to distinguish. Including HV, however, provides additional information, since OW areas on this image appear generally darker than sea ice in HV.

(23) Section 4.1: As already noted above (comment 15): How did you handle the computation of texture features in the zones along the borders between the image stripes that you indicated as white lines in 3b?

When the sliding window reaches the first element of image stripes (masked by nan value) the all 'masked' stripe values are skipped, and the computation is continued when sliding window passes away the whole stripe's "width". Further the masked stripes are excluded from the result matrixes of texture features.

(24) Section 4.2: Please see comments (19). From my point of view it is up to you whether you provide more details about methodological aspects in section 3.4 or in this section 4.2.

Please, see our answer for the comment 19.

The table that lists the experiments definitely belong to the methodology. The Results may refer to the number of experiment in the table.

(25) Page 9, lines 3-5, Fig. 4: The usefulness of the different texture features is not clear to me. When combining the intensities at HH and HV, calm OW is very well separated, but OW and ice partly overlap. The latter is also valid for the two graphs to the right, but here even calm OW overlaps with the ice. Please provide some convincing arguments why the texture features improve the water-ice separation? (See also comment 33 below).

The scatterplot of HH-pol energy vs. HV-pol correlation have been added.

Using only combination of HH and HV intensity, we can distinguish open water in some cases (Fig 1c below). The co-pol ratio was calculated, and then simple threshold was used to figure the OW area. Here OW was distinguished clearly with some artifacts on the other part of the image.

[Figure]

a)  HH-pol                              b) HV-pol                    c) hh\hv ratio & threshold

Figure 1.

But in the most cases it is insufficiently (Fig. 2c). If the only combination of the intensities at HH and HV will be applied in the classifier with preliminary training procedures, the result will not be satisfactory.

[Figure]

a)  HH-pol                              b) HV-pol                   c) hh\hv (ratio) & threshold

Figure 2.

We also provide some examples of other SVM, where all TF and only $\sigma_{HH}$ and $\sigma_{HV}$ were used for training process. Some statistics are provided. Please see our answer to comment 9 by Referee 2.

(26) Referring to Fig. 6, 7, 8: why is Fig. 5 not mentioned before them?
We have corrected this.

(27) Section 4.4: How was the comparison between classified radar image and Met. Norge ice charts carried out? Were the latter digitized to the pixel size of your radar images?

The comparison was carried out using pixel by pixel approach. The met.no ice charts were originally in digital form in stereographic projection. Our SAR-based classification results were subsampled and reprojected onto the coordinate system of the ice charts for further comparison.

(28) page 9 line 28-30: (a) "...ice charts WERE obtained manually..."; (b) what do you mean by "higher rate of thematic processing" in this context? I think that also ice charts from Met.Norway could include a higher level of details if the ice analyst delimits small-scale features. But this would require a longer time for ice chart production. What is the processing (computer) time needed when applying your algorithm compared to the semi-automated chart production by Met. Norge? This aspect is important and should be added to the discussion section 5.

(a) We corrected this.
(b) We removed these sentences. Our developed automated algorithm allows computing a classification result in less than 15 minutes. Based on our experience with visual classification we can conclude that it may take up to several hours by an ice expert to produce the ice chart for users. Thus, our semiautomatic algorithm is much more efficient than the manual one.

(29) Section 5.1: (a) The title refers only to the incidence angle correction, but the noise correction is also dealt with, at least in the last sentence.
We have corrected the title as: Significance of incidence angle variations and thermal noise reduction.

(30) page 10, lines 5-6: "Strong dependence of the backscatter on incidence angle in open water surface... is significantly higher than for sea ice". This statement is not true in any case, since the incidence angle sensitivity depends on the surface roughness: the smoother the surface, the larger the sensitivity (in particular close to the angle of specular reflection). Thin level ice can be very smooth and may show a larger sensitivity than a wind-roughened water surface.

[Figure]

[Figure]

*a)* Sea ice (coefficient = -0.29)          b) Open water rough (coefficient = -0.68  (- 0.76)
**Figure**. Angular correction of RS2 SAR image HH-channel along the whole swath a) sea ice b) open water).

The coefficients were obtained by the averaging of the derived angular dependencies of the backscatter for sea ice and open water rough from a series of RS2 SAR images in winter.

We also have the investigations for Envisat ASAR WS data. Angular dependencies of the backscatter for various sea ice types were derived from a series of Envisat ASAR WS images. They are shown in a Table below.

Table. Angular dependences of sigma0 for several sea ice types and rough open water with incidence angle increase for all images.

| Ice type | Changes of sea ice backscatter with incidence angle increase, dB/degree |
|---|---|
| Water surface | (-0.82) – (-1.05) |
| Grey ice | (-0.14) – (-0.20) |
| First-year ice | (-0.18) – (-0.42) |
| Multiyear ice | (-0.16) – (-0.24) |

**(31) (a) page 10, line 8**: what do you mean by "overestimated signatures"?
**(b)** Line 9: what do you mean by "ice cover has more reliable backscatter ranges for various ice types"? Since wind speed can be determined from radar measurements over open water, this means that also open water has clear defined ranges of backscattering intensity for a given wind speed and direction (except for OW patches
10 within the ice cover and at the ice margin where ocean surface wave interactions are more complicated).
The sentences have been rewritten as follows:
Coefficients for angular dependence of water covered areas are significantly influenced by wind conditions - with stronger wind intensity grows faster. Our observations show that angular dependence of sea ice is more stable regardless of wind or other conditions. Since the surface type is not known a
15 priori, we have to choose which angular correction to apply and the preference is given to the more reliable sea ice angular correction.

**(32) Section 5.2: (a)** You should make clear that in the first sentence you refer to the nine texture parameters given by equations 2-10. If you in addition use intensity and standard deviation, there are in total 22 parameters
20 considering both HH and HV (and not 20). In the following text you should make clear that you now refer to section 4.2, in which you selected 5 textural parameters + intensity + standard deviation for HH, and 4 textural parameters + intensity for HV, leaving in total 12 parameters.
**(b)** I do not understand the meaning of the two sentences from line 20 to line 23. E.g., what are "poor features"? What is a discriminant function in this context, and why is it needed? If all features are used, does
25 this mean that the classification accuracy is lower in any case? There is of course an optimal number and an optimal selection of textural features giving highest classification accuracy, but with, e. g., only 4 textural parameters you might theoretically still get better accuracies than with less parameters. The last 3 sentences of section 5.2 refer to the reduction of dimensionality but a direct link to the own processing described in section 4.2 is missing (e. g. the criteria for selecting some of the nine given textural features and excluding others).

(a) We agree that there is some confusion and corrected the text: one texture feature (cluster shade) was excluded from equations since actually we did not use it. Thus there are in total 20 parameters considering both HH and HV.
(b) 'Poor features' replaced by 'informationally poor features'
35 A sentences have been modified and moved from section 5.2 to section 3.4 :
A selection procedure is applied to limit a set of texture characteristics that provides a good classification with a small computational load. This procedure includes visual assessment of scatterplots comparing values of texture features in different combinations. Candidate texture features that provide

the best separation of classes are selected and others are discarded. The selection procedure also uses a set of training image regions to establish the set of features and its computation parameters providing the smallest classification error. In other words, we constrain the texture features number by the demanded balance considering the SAR image level of details, computation time and the optimal reliable class separation.

**(33) Section 5.3**: With this section I had considerable difficulties. **(a)** I suggest that you provide a table giving the function of each textural parameter (e. g. measure of local variations), and the interpretation of the respective parameter related to Fig 5.

10 **(b)** In this section you introduce alternative denotations for the textural parameters (e. g. energy – angular second moment; homogeneity – inverse difference moment etc). You should make this clear by expressing it like "Energy (also called 'Angular Second Moment')..." (after this, it is fine to go on with e. g. "Homogeneity or Inverse Difference Moment".

**(c)** page 10 line 30. I found that the "Energy" is the square root of the Angular Second Moment. Is
15 "repeatability" another denotation for energy? In Fig, 5i, the OW area, which appears homogenous in Fig. 3b, reveals a very low energy – is it a noise effect? The bright blue zones in the OW-area in Fig. 5i are due to the stripes in Fig 3b?

**(d)** page 11 sentence lines 6-7: I do not understand this sentence. The GLCM correlation function is calculated for a co-occurrence distance of 8 pixel? Fig. 5j: I do not understand why the correlation is very high in the
20 marginal ice zone and low in the more closed ice. I would expect it vice versa. The low correlation value over open water is again a noise effect?

**(e)** page 11, line 13: homogeneity – why is the homogeneity high in the marginal ice zone and low in the inner ice zone and over water (Fig. 5k)? Again I expected this vice versa.

**(f)** page 11, line 17: "...indicates a random mixture of scattering mechanism". I think here the entropy from the
25 entropy-alpha decomposition described by Cloude et al. (where entropy is indeed related to the character of the scattering mechanism) is mixed with the GLCM entropy, which does not give any information about the scattering characteristics from within one pixel but relations between neighboring pixels.

**(g)** page 11, lines 26-28 and lines 30-33, regarding Fig. 4: I do not see a clear separation between OW (dark blue) and sea ice (green), and the separation capability of the texture features seems to be worse than the one of the
30 intensities. See also comment 25 above.

The sub-sections 5.2 Number of texture features vs efficiency and 5.3 Meaning and value of texture features were merged and the text has been modified.

35 **(34) Section 5.4**, first paragraph: **(a)** it should be considered that the maps drawn by the ice analysts are a "smoothed version" of the ice cover variation.ns. In principle the ice analysts could also provide more detailed maps. However, I anticipate that an automated algorithm can do this much faster – see also comment 28 above.

**(b)** why didn't you use a better land mask?
40 **(c)** page 12 lines 17-20: how did you treat the beam boundaries when calculating the texture features (see also comment 23 above)?

**(d)** page 12, lines 32-33: why didn't you exclude cases in which the temporal difference between manual and automatic ice chart was large?

(a) Please, see our answer for the comment 28. We correct this.

(b) In fact landmask has high resolution, 250 m indeed. In order to mask incorrectly classified subimages along the shore, where land pixels are mixing with water or ice pixels, the landmask was extended to also cover the sea. We remove this sentence.

5    (c) The all 'masked' residuals are skipped on the TF calculation step. Then it is excluded from the result images – this is a part of the algorithm. This lost information does not caused significant harm in the image scale.

(d) We would like to test as much data as we can, and estimate the fully automated process – the SAR images were acquired and all of them were processed in the zone limited by the latitude and longitude
10   values. Actually we just tried to make sense that the accuracy can have higher value, and the potential of this classification technique is high for the Sentinel-1a data processing.

**Reviewer 2:**

15   **1. P2L26- Dual-polarization has several**

C: The major advantage offered by the HH and HV polarizations is that they are results from different backscattering mechanisms. σ HH is dominated by first-order scattering (direct backscatter with no multiple reflections), whereas σ HV is a result of multiple scattering (two or more reflections involving two or more scatterers)). Hence it easy to understand that the magnitude of σ HV is usually smaller than the magnitude σ HH
20   . The energy radiated towards the radar decreases significantly with each reflection. Rewrite also the text in P4 L11-14 keeping in mind the above explanation and the comment (9) by Referee 1. E.g. increased ice deformation increases also the amount of the multiple scattering as does the large volume scattering component from MYI.

We understand the mechanisms of HH\HV intensity difference. Here we mentioned the practical point
25   of view of this advantage. Please, see our answer for the comment 9 by Referee 1.

**2. P3L20-23**. C: Here the authors could also comment why they have not targeted to produce a sea ice concentration chart which would provide to the users and modelers more information than a binary open water/sea ice chart. As we can see from Fig. 8 the presented classification chart (8d) is not a good
30   approximation for an ice concentration chart (8b).

Ice types are required for many other users. Ice concentration is important but this parameter is calculated by other algorithms. This method aims estimate the several ice types. This paper is the first step to implement the algorithm where we distinguish the only ice \ water.

35   **3. P4L5-8.** C: Here you could add a remark that the classifier trained in the winter conditions is not ideal for the summer conditions.

We have clarified this in text as: In summer the contrast between backscatter intensities of the melted different ice types observed on the SAR image is diminished since surfaces become smoother and is dominated by meltwater. The intensities are reduced as well as contrast between ice and OW.

**4. P5 Sect. 3.1. and P8 Sect. 4.1**. C: I strongly support the suggestion (14) of Referee 1. Otherwise the statistical incidence angle compensation that you use is left unclear. In Sect. 4.1 you give just number (0.298) without

units and with wrong sign. I assume that you mean the slope coefficient −0.298 dB/1 o . I wonder why the magnitude of the coefficient is much larger than −0.196dB/1 o given in your 2013 paper for MYI or the coefficient −0.23dB/1 o estimated in Makynen et al. (2002) for FYI. It should be noted that in Makynen et al. (2002) the same targets in different images with different incidence angles are examined. If I have understood

5  correctly, in your 2013 paper you have studied targets which looked similar but appeared in different incidence angle ranges in the same image. What kind of procedure have you followed here? Is the steeper slope due only to a different sensor or do some geophysical factors contribute, like sometimes less than 100 % ice concentration in the test images?

Please, see our answer for the comment 14. We derived angular dependences of backscatter for sea ice

10  including different stages of development and deformation with a one degree step from calibrated RS2 SAR images at HH–polarization in winter period when the ice was observed continuously across the swath. There is no any geophysical factor since the images were taken from different parts of Arctic. We have corrected the slope coefficient according to the reviewer's remark.

15  **5. P6L2-7**. C: The referee 1 already commented this passage in the comments (16) and (17) which comments I support. My additional question is that what is the role of these subclasses in the classification scheme. Does SVM use them? If so, how have you selected all these subclasses as input to the classifier. Clarify the text please.

Please, see our answer for the comments 16, 17. The classification results were assessed using expert

20  knowledge (please, see also comment 12 by Referee 1). We use these classes for training. The procedure of SVM learning in our case involves the following steps: 1) RS2 ScanSAR images for training were selected and pre-processed. 2) These pre-processed images were used for texture features calculations. 3) The expert analysis of RS2 ScanSAR images includes the identification of plolygons with sea ice (3 ice types) and open water (calm and rough open water) delineation. These results were

25  collocated with texture characteristics matrixes (obtained after step 2) to get a number of training vectors. Selected dataset was applied for SVM training as inputs. As result three ice types and types of different open water were taken.

We have clarified this in Section 3.3 as:

30  These manual classification results were collocated with texture feature images (description provided in Sect. 3.4 and 4.2) to get a number of training vectors. For the final product the subclasses were merged into the main classes 'sea ice' and 'open water' since the similarities between the subclasses are too high for a reliable discrimination without additional data.

35  **6. P7L5-8**. C: An addition to the comment (19) of the Referee 1. Yu et al. 2012 (in your reference list) have applied to the feature selection problem "a forward feature search" which is identical to the forward stepwise selection in the regression analysis. The only difference is that in the feature selection the criterion is the classification accuracy instead of the criteria like AIC, BIC and many others used in the selection of the variables in the regression model.

40  Please see our answers to the comment 19 by Referee 1.
We have estimated obtained SVM versions with different TF combinations by testing it on our special "test" list. This test list was formed after the several algorithm training attempts and includes the most difficult situations for automated classification like: the open water areas with different surface features

caused by wind and currents represented a significant problem, several difficult situations in "transition zone" - area between a general ice massif and ice-free area, etc. The SAR images of the test list partly belong to the training list.

**7. P7**: Section 3.5. The description of the SVM is given in a very general level and the text is not well organized. The presentation should be more informative. You have many alternatives to detail your presentation. One is that you formulate the SVM as a solution to an optimization problem (e.g. Hastie et al, The Elements of Statistical Learning, available as a PDF file in the internet) and comment its properties from this point of view. Another approach is to treat the problem as Yu et al. 2012 (mentioned above) have done. In any case you must estimate some parameters when fitting the SVM in your data. Give the estimation method. When someone reads your text, he/she should get an idea what the SVM is and why you have chosen it. The equations are in this context necessary. The SVM gives only a binary classification result. Explain how you have generalized it into the case of three classes (like in Fig. 8c).

We have specified and corrected description of the SVM in the paper. We have not included the equations of the decision function and RBF kernel, since there are well-known and can be found in any SVM book or paper. The references are provided.

**8. P8L11**. C: Is the radar look direction in Fig. from right to left?

**9. P9L3-5**. C: I agree with Referee 1 (comment (25)) that Fig. 4 shows no increased discrimination ($\sigma_{HH}$, $\sigma_{HV}$) pair. When looking at Fig. 5 my ability with the texture features when compared to the subjective opinion is that Figs. 5a and 5h (corresponding to the HH and HV channels) provide the two best features. Show how the classification accuracy improves when you add texture features to the ($\sigma_{HH}$, $\sigma_{HV}$) pair. The sentence in P9L1-2 is not an argument.

We agree with this remark, and corrected the sentence: Texture characteristics provide a more complete delineation of surface parameters in addition to the raw backscatter signal, and increase the potential for ice and water separation.
The new scatterplot was added.

Please see our answer to comment (25) by Referee 1.
We produce several versions of SVM. The figures below present the classification results of SVM, when the whole set of TF were used for training (SVM1). Training of SVM2 was based on our selected TF (working version used for automated classification).

[Figure]

HH-polarization      HV-polarization      SVM1      SVM2
Figure 9.1: Ice edge map estimated from the RS2 image, 20130318.

[Figure]

| Original image | Angular corrected image | SVM1 | SVM2 |

Figure 9.2: Ice map estimated from the RS2 image, HH-polarization, 20130305.

5  To present the usefulness of TF, we have trained few SVM with only 2 input parameters - $\sigma_{HH}$ and $\sigma_{HV}$. Several SAR images were classified by SVM (SVM2) using in our automated algorithm presented in paper, and version of SVM with $\sigma_{HH}$ and $\sigma_{HV}$ in training procedure (SVM3) [please, see figures below].

[Figure]

| HH-polarization | SVM2 (selected TF) | SVM3 ($\sigma_{HH}$ and $\sigma_{HV}$) |

10  Figure 9.3: Ice map estimated from the image, HH-polarization, 20111122. Blue – open water, grey \ white – sea ice, green – land mask.

[Figure]

| a) SVM2 (selected TF) | b) SVM ($\sigma_{HH}$ and $\sigma_{HV}$) | c) SVM2 (selected TF) | d) SVM3 ($\sigma_{HH}$ and $\sigma_{HV}$) |

15  Figure 9.4: Ice map estimated from the image, a, b) 20130318; c, d) 20130305.
Blue – open water, grey \ white – sea ice, green – land mask.

The estimation of several RS2 images (accuracy, ice and ow errors) classified using SVM1, SVM2 and SVM3 are presented in Table below. The error matrixes based on pixel-by-pixel difference between

algorithm results and METno charts have been calculated for each RS2 image (listed in Table). Overall accuracies of OW and sea ice correspondence (Ov acc) and the impact of each class to the classification error (errors in ice and water classification – ow err, ice err, respectively) were computed.

| | RS2 image | SVM1 (all TF selected) | | | SVM2 (TF selected) | | | SVM3 (σHH and σHV) | | |
|---|---|---|---|---|---|---|---|---|---|---|
| | | Oc acc | ow err | ice err | Oc acc | ow err | ice err | Oc acc | ow err | ice err |
| 1 | RS2_20130206_ | 96.500 | 0.044 | 3.456 | 95.700 | 0.044 | 4.256 | 75.322 | 0.030 | 24.647 |
| 2 | RS2_20130206 | 96.409 | 0.000 | 3.291 | 94.309 | 0.000 | 5.691 | 61.043 | 0.000 | 38.957 |
| 3 | RS2_20130207_ | 95.6 | 0.58 | 3.784 | 95.889 | 0.327 | 3.784 | 69.442 | 1.337 | 29.221 |
| 4 | RS2_20130227_ | 95.889 | 0.327 | 3.784 | 93.045 | 0.626 | 6.329 | 56.986 | 0.159 | 42.856 |
| 5 | RS2_OK37130_ | 95.586 | 0.017 | 4.614 | 94.068 | 0.008 | 5.924 | 64.905 | 0.018 | 35.077 |
| 6 | RS2_20130313_ | 80.913 | 17.86 | 0.411 | 95.450 | 0.893 | 3.657 | 29.607 | 8.101 | 62.291 |
| 7 | RS2_20130314_ | 88.340 | 9.236 | 2.424 | 91.996 | 0.138 | 7.866 | 43.742 | 5.999 | 50.259 |
| 8 | RS2_20130401 | 85.428 | 12.57 | 1.998 | 88.691 | 0.371 | 10.937 | 32.271 | 4.294 | 63.435 |
| 9 | RS2_20130403_ | 93.142 | 3.860 | 2.998 | 93.239 | 0.446 | 6.314 | 26.798 | 6.862 | 66.340 |
| 10 | RS2_20130416_ | 84.024 | 15.199 | 0.778 | 94.544 | 1.459 | 3.997 | 37.667 | 8.474 | 53.859 |
| 11 | RS2_20130420_ | 69.692 | 27.336 | 1.516 | 93.031 | 2.183 | 4.786 | 26.037 | 8.710 | 65.253 |
| 12 | RS2_20130425_ | 87.011 | 12.441 | 0.548 | 94.251 | 0.246 | 5.504 | 45.462 | 8.027 | 46.511 |
| 13 | RS2_20130430_ | 59.403 | 37.231 | 3.366 | 93.658 | 0.443 | 5.899 | 29.038 | 28.24 | 42.720 |
| | wind stripes & eddies [current] | | | | | | | | | |
| 14 | RS2_20130301_ | 85.135 | 12.327 | 2.538 | 91.411 | 3.429 | 5.160 | 41.724 | 1.547 | 56.729 |
| 15 | RS2_20130301_ | 84.097 | 15.175 | 0.729 | 97.444 | 0.725 | 1.831 | 66.186 | 9.863 | 23.950 |
| 16 | RS2_20130304_ | 72.827 | 27.114 | 0.059 | 97.002 | 1.609 | 1.389 | 53.131 | 0.112 | 46.757 |
| 17 | RS2_20130305_ | 42.675 | 55.731 | 1.593 | 83.854 | 9.518 | 6.627 | 36.714 | 4.656 | 58.630 |
| 18 | RS2_20130305_ | 58.442 | 41.558 | 0.000 | 96.925 | 2.300 | 0.775 | 58.391 | 0.471 | 41.138 |
| 19 | RS2_20130306_ | 84.726 | 14.309 | 0.965 | 93.524 | 0.269 | 6.207 | 26.812 | 0.310 | 72.878 |
| 20 | RS2_20130319_ | 57.011 | 42.989 | 0.000 | 85.124 | 14.43 | 0.446 | 53.895 | 17.88 | 28.221 |
| 21 | RS2_20130408_ | 92.325 | 5.291 | 2.384 | 92.159 | 0.496 | 7.345 | 14.425 | 3.379 | 82.195 |
| 22 | RS2_20130408_ | 85.609 | 13.881 | 0.510 | 92.899 | 2.030 | 5.071 | 15.574 | 0.702 | 83.723 |

Correct detection of open water areas with different surface features caused by wind and currents represented a significant problem for automated classification that motivated us in our research of TF selection. Visual analysis (Fig. 9.1 – 9.2) and inspection of overall accuracies of SVM1 and SVM2 shows that although the sea ice delineation is more accurate (images 1-13 in Table), the optimal TF number can improve OW detection (the images from 14 to 23 in Table contain different the most difficult features for automate detection on the OW surface).

**10. P9L2**. ... methodology description... C: What did you mean by this? In the 2013 paper you selected all the features. Please clarify.
Please, see our answer for the comment 19 by Referee 1.

**11. Sect. 4.2. and Fig.4**. C: How have you normalized the features? As Fig. 4 shows the ranges of different textural features are highly variable. It would also be better if the normalized values of the textures (as in Fig. 5) would be used in Fig. 4. In any case the SVM requires that normalized feature values are used or the distance concept in the radial basis function is arbitrary.
All features were normalized for SVM training \ application using defined mean and standard deviation values of each TF from all training images. The subimages of figure 4 were corrected.

**12. Fig. 8**. In the figure caption: ...open water (ice concentration from 0 to 15%)... C: How have you identified such areas? The manual ice charts has the ice concentration classes: 0/10 -1/10, 1/10 -4/10 and so on. The class 0/10 -1.5/10 is missing.

In our previous calculations we used ice concentration data from OSI-SAF database, where the 15 % threshold was used for OW and ice separation. In this case for validation of automated classification result we use ice charts from METno with threshold amounts 0/10.
We agree with this remark and corrected this.

**13. P9L28-30.** C: I disagree with your conclusion that the SVM classification gives a more detailed ice cover map than the manual ice chart. If we inspect Fig. 8b we see how the sea area is divided into subareas with different ice concentrations. In Fig. 8e you have thrown away all this valuable information and forced the manual ice chart to a binary map. The comparison between the automated and manual chart that you have presented in the text is not fair. Please modify your text and assessment.

We understand that all ice services (not only met.no) in their manual interpretation can produce very detailed high quality ice charts with rather high accuracy of the ice situation. And in any case we were not going to dispute the valuable and accuracy of met.no charts since we use it as the best available information for validation. The text was modified.

**14. Sect. 5.2**. C: An addition to the comment (32) by the Referee 1. Do you have considered the principal components as a way to deal with the intercorrelation of the features and simultaneously reduce the dimensionality? If you have, why did you reject the principal component analysis.

We have tried this approach in our previous attempts of RS2 classification using Neural Networks. The PCA was applied to reduce the TF number in input vector. The visual analysis of classification results and the estimated accuracies were not so satisfactory.

**15. Sect. 5.3**. C: As the Referee 1 (the comment (33)) I struggled and often failed to understand your interpretations of the texture measures. This section had to be rewritten, e.g. following the guidelines given by the Referee 1. Just one addition. As far as I know, the only scattering mechanism one is able to measure from the dual-polarized HH +HV image is the depolarization ratio. In the decibel .scale the depolarization ratio is simply the difference σ HV -σ HH

 Also please see our answer to the comment 25 by Referee 1.

**16. P12L4-5.** C: I disagree with you due to the same reason as earlier. I think that for the models a sea ice concentration estimate at coarser resolution is a better option than knowledge of locations of small open water patches or leads.

Also please see our answer to comment 2 and 9.

**17. P12L6-8.** C: It is possible to derive a land mask from the MODIS data at resolution of 250 m. So the difference is not big compared to RS2 data, especially when we take into account that the resolution of the final product is 1.6 km (P5L6). Why do a MODIS based land mask underestimate the land area? I would expect that it might slightly overestimate it.

In fact landmask has high resolution, 250 indeed. In order to mask incorrectly classified subimages along the shore, where land pixels are mixing with water or ice pixels, the landmask was extended to also cover the sea.

5   18. P12L15. ...different structures on the water affected by wind and .. C: Eddies are not caused by winds. They are results of ocean currents. Write ...affected by wind and currents ..

Yes, of cause eddies are results of ocean currents. We agree with this remark and have corrected the text.

10   **Reviewer 3:**

**General Comments: 1. Too often are new image classification algorithms just assessed on a few test images and as a result are not very robust. A strength of this paper is that the algorithm the authors present is tested over a great number of images. In this regard, the classification results are very good over the annual cycle and while their Table nicely summarizes this result it would be useful to showcase the classification results in**

15   **more detail. Specifically, I think readers would like to actually see (visually) the performance of this algorithm during the summer melt or more difficult classifications – I know, I did after reading this paper. I suggest adding a few more examples or even a panel figure of classification comparisons with ice charts highlighting algorithm performance visually. They do not have to be perfect but for operations that does not matter – ice analysts want to see how the algorithm will perform in the most difficult conditions.**

20   Please see the figures for the comment 9 by Referee 2.

Yes, it is possible to add more examples, but we don't want to extend the number of figures and would like to maintain the existing layout of the paper.

**2. The English structure requires some serious attention. There are numerous passages that are difficult to follow or just do not make sense. I suggest a thorough English edit be required before publication.**

25   **3. The manuscript structure can be improved by combining the results (Section 4) and discussion (Section 5) sections. As it reads now, certain sub-sections of the discussion do not reference material created in the analysis which they should do (i.e. 5.1, 5.2). For example, the discussion on incidence angle has no reference to the correction the authors applied. Validation and Sources of Error can be easily compared and would make for a better read.**

30   **4. Perhaps more important than 2 and 3, it is not clear from the text (4.2/5.2) how the optimal texture combinations where chosen? I think this needs to be addressed in the methodology not the results. Nevertheless, this a remains a major problem that needs to be clarified.**

Please, see our answer for the comment 19 by Referee 1.

35   **Overall, I think the algorithm presented in this paper is worthy of publication when the comments outlined above and below are taken into consideration.**

**P1L22: I don't think exploring is the correct word. Perhaps quantifying?**

We rewrote the sentence.

**P1L24: regions not region**

We have corrected this sentence.

P1L25: such as ERS-1/2
We have corrected this sentence.

P1:30: that extend operational utility.
We have corrected this sentence.

P2L1: The objective of sea ice classification is to identify sea types and open water. You do not need "based on" unless you are going to mention everything taken into consideration.
We have corrected this sentence.

P2L5: That is basically all factors. Why not just say discriminating between open water that is wind roughened and sea ice is difficult?
We have corrected this sentence.

P3L4: The CIS did not developed MAGIC. To my knowledge it was developed by Dr. Clausi at the University of Waterloo.
Yes, the MAGIC (or MAGSIC) was developed in University of Waterloo by the MAGIC research group (Clausi, Maillard, Deng etc). MAGSIC had shown promising results and is being encouraged by CIS. Now MAGSIC is a modular system being developed as an operational tool to be inserted into CIS operations. We agree with this remark and corrected the text.

P3L20: The goal is not to extend ENVISAT single polarization, it is simply to utilize dual polarization data for ice classification.
We agree with this remark and corrected this sentence.

P3L30: ice conditions.
This section contains some technical details of utilized SAR data and the study area as well as ice conditions on the winter images are described. So we think that "DATA" is more common title for this section.

P4L10-20: No need to describe what HV is. Start with: The HV channel...but this is a difficult passage to follow on the physics as to why HV is darker than HH in RADARSAT-2. I suspect English is the root cause. Revise.
Please, see our answer for the comment 9, 10 by Referee 1.

P4L22-33: It would be better if the methodology was written out in paragraph form rather than numbered points. You can still include numbers (i.e. i, ii, iii, etc) in the text.
We have decided to keep this style.

P5L13: Why not just simply state that the imagery was normalized to 35 degrees and move on? IA correction does not require a separate sub-heading.
We followed to the comment 14 by Referee 1. Also please see our answer to the comment 4 by Referee 2.

P5L30: Unclear what is meant by Manual classification has be done. . .? Did the author's manually classify the imagery? It is unclear what is trying to be communicate in this sub-heading.
We have clarified this section.

P6L10: Are all the texture features used in the classification or just some? How are certain ones selected over others? This needs to be clear in the text. See General Comment 4.
Please, see our answer for the comment 19 by Referee 1.

P8L5: Is there a website link to the MET ice charts? Do they use RADARSAT-2 imagery? If they do not, this should be mentioned as they are an independent source for comparison.
We have added a website link to MET ice charts. We mentioned in text that they use also high resolution SAR images (RS2 and Sentinel-1A) in the analysis.

P8:10: Why is figure 6 being introduced before the other figures?
We have corrected this.

P13L6: developed, not proposed.
But we already developed the algorithm and now proposed the algorithm description in this paper.

Figure 2: Needs some latitude/longitude information for reference
Done.
Figure 3: Needs geography similar to Figure 2. The line graphs need to include axis labels and the font needs to be bigger.
Done.
Figure 6. Labels a) and b) are not included on the image
Done.

[revised manuscript text omitted]

---

## Referee Report (RR1)

[referee-annotated manuscript omitted]

---

## Author Response (AR2)

**RESPONSE TO REVIEWER COMMENTS**

Dear Referees,

Thank you very much for your constructive suggestions and improvements. A marked-up manuscript version shows the changes made in the revised paper.

Answers (in BLUE) to specific comments:

**Reviewer #1:**

The authors have addressed most of my concerns but one issue still remains:

The presentation of the methodology still needs improvement as the authors have not addressed my concerns. The scientific description of the methods needs to be clear and the bullet points are not appropriate because they are then followed sub-sections of Incidence angle correction, thermal noise correction, etc. that is not logical in terms of flow. For example, Step 1 describes items in the following sub-sections so it falls apart almost immediately. Section 3 needs to be revised in a logical, orderly manner which can be done easily: remove the text before 3.1 and then just describe the approach.

We rewrote the Section 3 according to Referee's suggestion.

**Page 5, Line 19:** Remove text: "The next step will be to adapt and apply the algorithm for classification of several ice types using Sentinel-1A/B as the main data source." This is not relevant to this study and does not belong in the Abstract.

Done.

**Page 15 Line 29:** Add a reference to the corrected image (Figure 3) to support your discussion.

Done.

**Reviewer #3:**

Thank you very much for your suggestions and we have agreed with them with a few exceptions: P7L21 and P8L4-5.

**P10L18**: what do you mean by "undertaken"?e. g. ...reliable and of good quality ?

We agree with your comment. We rewrote this sentence.

**P12L14-16:**" ...The basic SVM takes a set of input data (several "attributes", i.e. the features or observed variables) and predicts (i.e. the class labels) for each given input ...." predicts what?

We have corrected this sentence as following: The basic SVM takes a set of input data (several "attributes", i.e. the features) and predicts the output (i.e. the class labels) for each given input, making it a non-probabilistic classifier.

**P12L27:** "SVM models derived from LIBSVM software, which is applied in scikit-learn." This is not a correct sentence. What do you want to say?

SVM models implementation in scikit-learn is based on LIBSVM.

**P14L24-25:** "…Including more texture features for both channels was tested but found not to improve the information content." - what happens if you take away some of them?

We just mentioned that it is not necessary to include many (all) of TF since it does not improve classification results and sometimes might lead to results of lower quality. TF number reduction also reduces computational burden.

**P15L27-29:** "..The coefficients for the angular dependence of water covered areas are significantly influenced by wind conditions - with stronger wind intensity grows faster." … with stronger wind, the backscattering intensity grows faster (is this what you meant? But this is wrong, since the backscattering intensity over open water saturates at high wind speeds! Please check.)

We already had the similar question before. We apologize may be we don't understand clearly the question, but just repeat our answer.

[Figure]

[Figure]

*a)* Sea ice (coefficient = -0.29)          b) Open water rough (coefficient = -0.68  (- 0.76)

Figure. Angular correction of RS2 SAR image HH-channel along the whole swath a) sea ice b) open water).

The coefficients were obtained by the averaging of the derived angular dependencies of the backscatter for sea ice and open water rough from a series of RS2 SAR images in winter.

We also have the investigations for Envisat ASAR WS data. Angular dependencies of the backscatter for various sea ice types were derived from a series of Envisat ASAR WS images. They are shown in a Table below.

Table. Angular dependences of sigma0 for several sea ice types and rough open water with incidence angle increase for all images.

| Ice type | Changes of sea ice backscatter with incidence angle increase, dB/degree |
|---|---|
| Water surface | (-0.82) – (-1.05) |
| Grey ice | (-0.14) – (-0.20) |
| First-year ice | (-0.18) – (-0.42) |
| Multiyear ice | (-0.16) – (-0.24) |

**P16L13-15:** " …then the informationally poor features have to be compensated by using complicated discrimination function and can lead to increased classification confusion." -  I do not understand what you mean by this part of the sentence. Are you talking about the effect that too many correlated parameters have on the performance of the SVM?

Yes, the same idea as was noted in the previous answer P15L24-25.

**P17L9**: " … due to its very smooth ice signature." The MIZ does not have a smooth signature. It often shows high-frequency variations of the intensities due to the mixture of different ice types and open water.

We defined MIZ as rather narrow area between OW and ice, which consist of, in general, the same ice type as the main closed ice area, broken ice and some more younger ice with mixture of open water, but we can not distinguish the ice types \ water inside this area. This area is particularly difficult to classify

automatically due to the impossibility to separate the highly variable signatures on a small scale due to the "smoothing" of the different signatures. We have removed this sentence.

**P17L28:** don't you mean "increases"? When you reclassify Met Norway charts, the agreement with your results improves.

We mean reduces. The classes disagree between the MET Norway manual ice charts and our classification results. In other words, our attempt to take into account the 10 %  threshold for distinguishing between OW and ice used on MET Norway ice charts makes impossible to do an accurate estimation of the ice conditions in the SAR images and their detailed comparison. In the 0-10 % zone indicated as "open water" on met.no charts we can have some ice features on the SAR image and in our classification result, but the comparison will give an error of our classification.

**P17L29-35**: Why didn't you exclude images with too large temporal differences to the MET Norway charts?

We would like to test as much data as we can, and estimate the fully automated process – the SAR images were acquired and all of them were processed in the zone limited by the latitude and longitude values. Actually we just tried to prove that the accuracy can have higher value, and the potential of this classification technique is high for the Sentinel-1A data processing.

**P18L10:**" …by a wave-like modulation of the image intensity in azimuth direction … " -don't you mean "range"? That is what I conclude from Fig. 2. Please indicate range and azimuth direction in Fig. 2.

Yes, we mean range direction.

**Figure 2**: Please indicate range and azimuth direction. If range is horizontal from left to right, then I see noise floor modulations along range, but not along track (?)

We have indicated range and azimuth directions.

**Figure 3**: " …The blue curve shows σ° value profile of the raw HV channel image over the horizontal line … " - over which horizontal line?

The caption was improved.

**Figure 4**: But what are texture features 0 and 10 - 19 ?

The caption was improved.

[revised manuscript text omitted]